



# Measurement report: Quantifying source contribution and radiative forcing of fossil fuel and biomass burning black carbon aerosol in the southeastern margin of Tibetan Plateau

Huikun Liu[1,2,3], Qiyuan Wang[1,2,3,4*], Li Xing[5], Yong Zhang[2], Ting Zhang,[2] Weikang Ran[2], Junji Cao[1,2,3,4*]

[1]State Key Laboratory of Loess and Quaternary Geology, Institute of Earth Environment, Chinese Academy of Sciences, Xi'an, 710061, China
[2]Key Laboratory of Aerosol Chemistry and Physics, Institute of Earth Environment, Chinese Academy of Sciences, Xi'an, 710061, China
[3]University of Chinese Academy of Sciences, Beijing, 100049, China
[4]CAS Center for Excellence in Quaternary Science and Global Change, Xi'an, 710061, China
[5]School of Geography and Tourism, Shaanxi Normal University, Xi'an, 710119, China

*Correspondence to*: Qiyuan Wang (wangqy@ieecas.cn) and Junji Cao (cao@loess.llqg.ac.cn)

**Abstract.**

Black carbon (BC) aerosol plays a vital role in disturbing the balance of ecosystem and climate stability of Tibetan Plateau (TP). An intensive campaign was carried out from 14th March to 12th May 2018 in the southeastern margin of TP to investigate the sources of BC and their radiative effects. To do so, an improved 'aethalometer model' was used to distinguish and apportion BC into fossil fuel combustion source and biomass burning source. To minimize the uncertainty associated with the 'aethalometer model', a receptor model coupling multi-wavelength absorption with chemical species was used to retrieve the site-dependent Ångström exponent (AAE) and BC mass absorption cross-section (MAC). The results show that the AAEs and BC MACs at wavelength of 880 nm were 0.9 and 12.3 $m^2$ $g^{-1}$ for fossil fuel source and 1.7 and 10.4 $m^2$ $g^{-1}$ for biomass burning, respectively. Based on these parameters, the fossil fuel source-related BC ($BC_{fossil}$) was estimated 43% of the total BC and the rest 57% was from biomass burning ($BC_{biomass}$) during the campaign. The results from a regional chemical dynamical model reveal that high $BC_{biomass}$ was contributed from the northeastern India and northern Burma, and the Southeast Asia can explain 40% of $BC_{biomass}$. The high $BC_{fossil}$ was mainly identified from the southeast of sampling site. A radiative transfer model estimated that the atmospheric directive radiative forcing of BC was +4.6 ± 2.4 W $m^{-2}$ on average, including +2.5 ± 1.8 W $m^{-2}$ from $BC_{biomass}$, and +2.1 ± 0.9 W $m^{-2}$ from $BC_{fossil}$,



which correspond to and heating rates of $0.07 \pm 0.05$ and $0.06 \pm 0.02$ K day$^{-1}$, respectively. Our study will be useful for improving our understanding in BC sources on the TP and their climatic effect.

## 1 Introduction

The Tibetan Plateau (TP) is considered as a regulator of climate change in the northern hemisphere and plays a crucial role in global ecosystem and climate stability (Yang et al., 2014). It covers one of the largest ice masses of the Earth system, which is known as the water tower of Asia (Immerzeel et al., 2010). The glaciers on the TP recently shows are rapidly retreating, which could lead to adverse effects on Asian hydrological cycle and the formation of the Asian monsoon (Wu et al., 2015). Although the causes of glacier melting are complex, the increasing black carbon (BC) loading on the TP has an inescapable role in this process (Yang et al., 2015). The BC deposited on glaciers increases ice melting by the enhanced radiation absorption on the surface, which further influences the mass balance of glaciers (Ming et al., 2009). The main paths for BC in glaciers are scavenged from atmosphere through dry and wet deposition (Ménégoz et al., 2014). Thus, exploring the source features of atmospheric BC on the TP is critical for further understanding its environmental and climatic effects.

As well known, BC is a byproduct from incomplete combustion process of carbon contained fuels (e.g., fossil fuel and biomass) (Bond et al., 2013). Currently, the majority of the atmospheric BC studies on the TP focuses on characterizing the spatial and temporal distributions of bulk BC aerosol, and  its regional sources (e.g., Rai et al., 2019; Wang et al., 2015). However, few studies shed light on the quantification of BC from different sources and their regional impacts. Owing to the various physicochemical characteristics, the climatic effects of BC produced by different sources could be divergent. Thus, it is important to understand the sources of BC on the TP. Among those limited available studies, they mainly focus on two aspects. One is obtaining the contribution fractions of different regions to the source-specific BC on the TP using modeling methods (Zhang et al., 2015); the other is based on the field observations to apportion BC into different sources through a certain data analytical method. The former approach has advantage in understanding the influence of BC regional transport to TP, but the result strongly depends on the accuracy of the emission inventory of different BC sources. The latter one can be achieved by multiple methods, for example, the filter-based approaches of isotope analysis (e.g., $\Delta^{14}$C/$\delta^{13}$C, Li et al.,



2016). The isotope approach has a good accuracy in BC source apportionment, but the sampling time is usually as long as 24h or even longer to ensure the enough sample on filter to meet the requirement of detection limits of analyzer. Thus, this method is limited by its low time resolution and fails to capture the accurate time of pollution events occurred on the TP.

Online methods, on the contrary, are advantageous to capturing pollution events because of its high time resolution feature. Among them, a multi-wavelength optical method called aethalometer model has been widely used in two-source BC apportionment globally (e.g., Herich et al., 2011; Zhu et al., 2017; Rajesh and Ramachandran, 2018). Although 'aethalometer model' can apportion BC with a high time resolution, its robustness depends on the selection of absorption Ångström exponent (AAE) and BC mass absorption

cross-section (MAC) used in the model. In current studies, most of them cited source-specific AAEs from previous publications and assumed the same BC MACs regardless of its difference in sources (e.g., Healy et al.; 2017;Zhu et al., 2017; Martinsson et al., 2017; Rajesh and Ramachandran, 2018; Dumka et al., 2018; Liu et al., 2018; Helin et al., 2018). This introduces great uncertainties, because even for the same type of emission source, the AAE and MAC can vary with specific fuel subtypes and combustion

conditions (Wang et al., 2018; Tian et al., 2019). To improve the accuracy, several studies tried to constrain the AAE values by a comparison between different source-generated BC and the source markers (e.g., levoglucosan) or by the $^{14}$C method (Sandradewi et al., 2008; Helin et al., 2018; Zotter et al., 2017). However, several studies have shown that the 'aethalometer model' is not as sensitive to the AAE variation of biomass burning source as to that  of fossil burning source (Dumka et al., 2018; Titos et al.,

2017), thus, the comparison method may be less effective to nail down the accurate source-specific AAE. In addition, all the 'aethalometer model' users ignore the contribution of light absorption formed by the secondary processes which may account for a substantial fraction at short wavelengths (e.g., 370 and 470 nm) (Wang et al., 2019a; 2019b). Consequently, this will lead to an overestimation of BC source apportionment results.

BC is recognized as the second largest anthropogenic warming agent globally after carbon dioxide, and its radiative effect varies largely with different areas (Bond et al., 2013). Currently, regional modeling and observation-based methods have been used in assessing BC radiative forcing on the TP. In this regard, most studies concentrate on the areas of longitude between 72 – 86° (e.g., Rajesh and Ramachandran,





2018; Bhat et al., 2017; Dumka et al., 2013; Singh et al., 2018). These studies focus on the radiative effect caused by bulk BC aerosol and some of them indicate that BC can account for 55 – 70% of the composite aerosol atmospheric radiative forcing (Panicker et al., 2010; Srivastava et al., 2012; Sreekanth et al., 2007). As summarized above, owing to the lack of studies regarding BC source apportionment on the TP, to our knowledge, there is no literature referring to the impacts of source-specific BC on the radiative forcing at present.

In this study, a receptor model coupling with multi-wavelength absorption and chemical species was utilized to retrieve the site-dependent AAEs and BC MACs, and then an improved aethalometer model was used to determine the mass portions of BC aerosol from fossil fuel combustion source and biomass burning on the TP. Regional transport of source-specific BC was further explored by models. Finally, the radiative effect caused by BC from different sources were evaluated using a radiative transfer model. Our study provides insights into the BC sources on the TP and their impacts on regional climate.

## 2 Methodology

### 2.1 Sampling site

Intensive observation from 14 March to 13 May 2018 were conducted on the rooftop of. a 10m height building above the ground (3260 m a. s.l.) in the Lijiang Astronomical Station, Chinese Academy of Sciences (100.03°E, 26.70°N), which was located in Gaomeigu County, Yunnan Province, China (Fig. S1). The location connects high altitude TP with low altitude Yungui Plateau forming a transportation channel for pollutants from Southeast Asia (Wang et al., 2019a). The population of Gaomeigu County is small, and limited anthropogenic activities were found around the sampling site.

### 2.2 Online and offline measurements

Aerosol light absorption coefficient ($b_{abs}$) was retrieved by model AE33 aethalometer (Magee Scientific, Berkeley, CA, USA) at multiple wavelengths covering from near-ultraviolet to near-infrared. The sampled particles were selected with a $PM_{2.5}$ cuff-off inlet (SCC 1.829, BGI Inc. USA) and dried with a Nafion® dryer (MD-700-24S-3; Perma Pure, Inc., Lakewood, NJ, USA) using a flowrate of 5 L min⁻¹. Detailed operation principle of the AE33 aethalometer has been elaborated in Drinovec et al. (2015).



Briefly, light at λ = 370, 470, 520, 590, 660, 880, and 950 nm from diodes were utilized to irradiate the filter deposition spot, and the produced light attenuation is detected by optical detectors. Non-linear loading and filter matrix scattering effects are common issues of filter-based absorption measurements (Coen et al., 2009). The former issue was resolved using a dual-spot compensation technique embedded in AE33 aethalometer while a factor of 2.14 was used to correct the latter issue caused by quartz filters (Drinovec et al., 2015).

A photoacoustic exinctionmeter (PAX, Droplet Measurement Technology, Boulder, CO, USA) was used to determine the aerosol light scattering coefficient ($b_{scat}$) and $b_{abs}$ as well as obtaining the single scattering albedo (SSA = $b_{scat}/(b_{scat}+b_{abs})$) at wavelength of 532 nm. The $b_{scat}$ was measured using a wide-angle (5 – 175 degree) integrating reciprocal nephelometer in the scattering chamber; and the $b_{abs}$ was measured simultaneously with an intracavity photoacoustic technique in the acoustic chamber. More detailed description of PAX can be found in Carrico et al. (2018). During the campaign, the $b_{scat}$ and $b_{abs}$ measurements were calibrated by different concentrations of ammonium sulphate and freshly-generated propane soot, respectively. Detailed calibration procedure has been elaborated in Wang et al. (2018).

Twenty-four-hour $PM_{2.5}$ samples were collected from 10:00 local time to 10:00 the next day on quartz fibre filters (8 × 10 inches, QM/A™; Whatman, Middlesex, UK) by a high-volume sampler (Tisch Environmental, Inc., USA) with an operating flowrate of 1.05 $m^3$ $min^{-1}$. The blank filters before sampling and the loaded filters after sampling were weighted to obtain PM2.5 mass in a thermostatic room using a Sartorius MC5 electronic microbalance (Sartorius, Göttingen, Germany). The sampled filters were well-stored in a refrigerator at -4°C to reduce the loss of volatile substances prior to chemical analysis. The chemical species involved carbonaceous matter (i.e., organic carbon (OC) and elemental carbon (EC)), water-soluble inorganic ions (i.e., $K^+$), and inorganic elements (i.e., S. Ca, Ti, Mn, Fe, Cu, As, Br, Pb, Zn), which were determined by thermal/optical carbon analyser, ion chromatograph, and energy-dispersive X-ray fluorescence spectrometry, respectively. Furthermore, organic markers of levoglucosan and benzothiazolone were measured using an ion chromatograph and a high-performance liquid chromatography (HPLC), respectively. More detailed information regarding above measurements is shown in Text S1 in the supporting information.



## 2.3 BC source apportionment

The aethalometer model (Sandradewi et. al., 2008) was improved by considering the $b_{abs}$ caused by the secondary formation and soil dust. Therefore, the $b_{abs}$ of the BC generated from fossil fuel combustion and biomass burning were calculated as follows:

$$\frac{b_{abs}(370)_{fossil}}{b_{abs}(880)_{fossil}} = \left(\frac{370}{880}\right)^{-AAE_{fossil}} \tag{1}$$

$$\frac{b_{abs}(370)_{biomass}}{b_{abs}(880)_{biomass}} = \left(\frac{370}{880}\right)^{-AAE_{biomass}} \tag{2}$$

$$b_{abs}(880) = b_{abs}(880)_{fossil} + b_{abs}(880)_{biomass} \tag{3}$$

$$b_{abs}(370) = b_{abs}(370)_{fossil} + b_{abs}(370)_{biomass} + b_{abs}(370)_{secondary} + b_{abs}(370)_{dust} \tag{4}$$

where $AAE_{fossil}$ and $AAE_{biomass}$ are the AAEs corresponding to fossil fuel contribution and biomass burning emissions, which were derived from an optical source apportionment method as discussed in section 3.1; $b_{abs}(370)$ and $b_{abs}(880)$ are the measured $b_{abs}$ at $\lambda = 370$ and 880 nm, respectively; $b_{abs}(370)_{fossil}$ and $b_{abs}(880)_{fossil}$ are $b_{abs}$ of fossil fuel BC at $\lambda = 370$ and 880 nm, respectively; $b_{abs}(370)_{biomass}$ and $b_{abs}(880)_{biomass}$ are $b_{abs}$ of biomass-burning BC at $\lambda = 370$ and 880 nm, respectively; $b_{abs}(370)_{secondary}$ represents $b_{abs}$ caused by secondary processes at $\lambda = 370$; $b_{abs}(370)_{dust}$ represents $b_{abs}$ associated with soil dust at $\lambda = 370$, which was calculated in section 3.1. In this study, a BC-tracer method coupled with a minimum $R$-squared approach was used to retrieve the $b_{abs}(370)_{secondary}$ (Wang et al., 2019a):

$$b_{abs}(370)_{secondary} = b_{abs}(370) - \left(\frac{b_{abs}(370)}{BC}\right)_{pri} \times [BC] \tag{5}$$

where $\left(\frac{b_{abs}(370)}{BC}\right)_{pri}$ represents the $b_{abs}(370)$ associated with BC in the primary emissions(in unit of $m^2\ g^{-1}$), and $[BC]$ denotes the mass concentration of BC in the atmosphere (in unit of $\mu g\ m^{-3}$), which was calculated with $b_{abs}(880)$ and EC loading. Detailed description of the $b_{abs}(370)_{secondary}$ calculation can be found in Wang et al. (2019a).

After calculating $b_{abs}(880)_{fossil}$ and $b_{abs}(880)_{biomass}$ based on Eqs. (1) – (5), the mass concentrations of BC from fossil fuel combustion and biomass burning ($BC_{fossil}$ and $BC_{biomass}$, respectively) were estimated as follows:





$$BC_{fossil} = \frac{b_{abs}(880)_{fossil}}{MAC_{BC}(880)_{fossil}} \tag{6}$$

$$BC_{biomass} = \frac{b_{abs}(880)_{biomass}}{MAC_{BC}(880)_{biomass}} \tag{7}$$

where $MAC_{BC}(880)_{fossil}$ and $MAC_{BC}(880)_{biomass}$ are the MAC of BC generated from fossil fuel combustion and biomass burning, respectively, which were retrieved from optical source apportionment as discussed in section 3.1.

## 2.4 Optical source apportionment

The optical source apportionment was realized via a positive matrix factorization (PMF) model. The fundamental principle of PMF is to resolve the chemical mass balance by separating data matrix into factor contributions and factor profiles as follows:

$$X_{ij} = \sum_{k=1}^{p} g_{ik}f_{kj} + e_{ij} \tag{8}$$

where $X_{ij}$ represents the input matrix elements; p is the number of sources; $g_{ik}$ is the source contribution of the $k$th factor to the $i$th sample; $f_{kj}$ is the factor profile of $j$th species in the $k$th factor; and $e_{ij}$ is representative of the residual. $g_{ik}$ and $f_{kj}$ are non-negative. The two matrices are resolved by minimizing the sum of squares of the normalized residuals as follows:

$$Q = \sum_{i=1}^{n} \sum_{j=0}^{n} \left[\frac{e_{ij}}{u_{ij}}\right]^2 \tag{9}$$

where $Q$ represents the object function; and $u_{ij}$ denotes the uncertainties of $X_{ij}$. The PMF version 5.0 (PMF5.0) was utilized, and the optical parameters (primary $b_{abs}$ at different wavelengths) and chemical species (including carbonaceous matter, inorganic elements, $K^+$, and organic markers) were both used as model inputs to perform the optical source apportionment.

## 2.5 Trajectory-related analysis

To determine the influences of BC regional transport to Gaomeigu, cluster analysis was based on the hourly three-day backward trajectories at 500 m above the ground level using the Hybrid Single-Particle Lagrangian Integrated Trajectory model (Draxler and Hess, 1998). Since we focused on differentiating




and clustering the main directions of coming trajectories, an angle-oriented distance definition was adopted in the cluster analysis. The detailed description of the cluster analysis can be found in Wang et al. (2018). To distinguish the pollution direction, the trajectory with BC mass concentration over 75th percentile was referred as a polluted trajectory, otherwise was a clean one.

The potential source contribution function (PSCF) was further used to identify the likely pollution regions that influenced BC loadings at Gaomeigu based on the backward trajectories. The geographic region covered by the trajectories was overlapped by a of $0.5 \times 0.5°(i, j)$ grid layer. The PSCF value of each grid was calculated as follows:

$$PSCF_{ij} = \frac{m_{ij}}{n_{ij}} \tag{10}$$

where $m_{ij}$ is the number of the endpoints associated with BC mass concentration higher than the set criterion; and $n_{ij}$ is the total endpoints of the $ij$th cell. To improve the resolution of PSCF source identifications (Cheng and Lin, 2001), the 75th percentile of each source's BC mass concentration was set as the pollution criterion (i.e., 0.6 µg m$^{-3}$ for BC$_{biomass}$ and 0.45 µg m$^{-3}$ for BC$_{fossil}$). Further, an arbitrary weighting factor ($w_{ij}$) was applied to different $n_{ij}$ ranges to reduce the uncertainty caused by the small $n_{ij}$

(Polissar et al., 1999), and it was defined as follows (Polissar et al., 2001):

$$W_{ij} = \begin{cases} 1 & 80 < N_{ij} \\ 0.7 & 20 < N_{ij} \le 80 \\ 0.42 & 10 < N_{ij} \le 20 \\ 0.05 & N_{ij} \le 10 \end{cases} \tag{11}$$

## 2.6 Regional chemical dynamical model

The Weather Research and Forecasting model coupled with chemistry (WRF-Chem) model was used to quantify the biomass-burning contribution of Southeast Asian to BC mass at Gaomeigu. More detailed

descriptions of the model configurations are described in our previous publication (Xing et al., 2020). Briefly, the model resolution was 3 km × 3 km, and there were 320 grids in each of the latitude and longitude. The domain was concentrated in the southwest of China, South Asia, and Southeast Asia, with the central location at 100.03°E, 26.70°N. Thirty-five vertical layers has been set in the model from the



ground surface to 50hPa. A BC emission inventory of 0.25° × 0.25°spatial resolution was used in model, including industry, power, transportation, and residential sources (e.g., fossil fuel and biofuel), which was based on the Asian anthropogenic emission inventory (that is MIX) for the year of 2010 (Li et al., 2017). The FINN fire inventory was used for the biomass-burning emission during the simulation.

## 2.7 Estimations of direct radiative forcing and heating rate

The direct radiative forcing (DRF) of source-specific BC was estimated by the widely used Santa Barbara DISORT Atmospheric Radiative Transfer (SBDART) model. The detailed description of SBDART model is elaborated in Ricchiazzi and Yang, (1998). The important input parameters include aerosol optical depth (AOD), extinction coefficient, SSA, asymmetric parameter (ASP), and visibility. These aerosol optical parameters were estimated by Optical Property of Aerosol and Cloud (OPAC) model using Mie theory (Hess et al., 1998). The measured BC, water-soluble matter (including measured water-soluble inorganic ions and water-soluble organic matter that assumes accounting for 79% of OC loading on the TP, Xu et al., 2015), and water insoluble matter (calculated by the mass concertation of $PM_{2.5}$ minus BC and water-soluble matter) were used in the OPAC model to retrieve the number concentrations of these matters which were then further used to obtain the optical parameters at the nearest observed relative humidity. These number concentrations were tuned till the modelled $b_{scat}$, $b_{abs}$, SSA were close to the ones measured by PAX, with a difference within ±5% (Srivastava et al., 2012) (see Table S1). The underlying assumption is when the modelled $b_{scat}$, $b_{abs}$ are very close to their measured counterparts, the rest derived optical parameters are assumed reasonable for the measured aerosols. This assumption has been widely accepted in previous studies (Dumka et al., 2018; Panicker et al., 2010; Rajesh et al., 2018). Finally, the DRF induced by source-specific BC alone (or $PM_{2.5}$) at the surface atmosphere (SUF) and the top of the atmosphere (TOA) were estimated by the difference in the net flux with and without BC (or $PM_{2.5}$) under cloud-free conditions.

Atmospheric forcing (DRF at TOA subtracts DRF at SUF) leads to solar heating rate change, which can be calculated as follows (Ramachandran and Kedia, 2010):

$$\frac{\partial T}{\partial t} = \frac{g}{C_p} \times \frac{\Delta F_{ATM}}{\Delta P} \tag{12}$$





where $\frac{\partial T}{\partial t}$ is the heating rate (K day$^{-1}$); $g$ is the acceleration due to gravity (9.8 m s$^{-2}$); $C_p$ is the specific heat capacity of air at constant pressure, $\Delta F_{ATM}$ is the atmospheric forcing; and $\Delta P$ is the atmospheric pressure between the ground and 3 km above.

## 3 Results and discussion

### 3.1 Source-dependent AAEs and MACs

Based on the optical source apportionment, four sources were identified contributing to primary $b_{abs}(\lambda)$ (Fig. 1). The simulated primary $b_{abs}$ ($\lambda$) at different wavelengths all correlated well (r = 0.96 – 0.97, p < 0.01, Fig. S2) with those model input values, indicating a good reproducibility of PMF5.0. As shown in Fig. 1, the first source factor exhibited high contributions of K$^+$ (90%), levoglucosan (60%), and primary $b_{abs}(\lambda)$ (45 – 64%) as well as moderate loadings of OC (38%) and EC (47%). The K$^+$ and levoglucosan are widely used markers for biomass burning (Urban et al., 2012). Owing to the presence of BrC, the absorption fraction increased toward to short wavelengths, which is consistent with the optical feature of biomass burning (Forello et al., 2019). Therefore, this source factor was assigned to biomass burning. Based on the contributions of biomass burning to $b_{abs}(370)$ and $b_{abs}(880)$, the $AAE_{biomass}$ was estimated to be 1.7, which is within a relative boarder range of $AAE_{biomass}$ (1.2 – 3.5) reported by previous studies obtained via comparison with other measurements (e.g., $\Delta^{14}C$ and organic tracers) in the atmosphere (Sandradewi et al., 2008; Helin et al., 2018; Harrison et al., 2012; Zotter et al., 2017). The estimated average $MAC_{BC}(880)_{biomass}$ was 10.4 m$^2$ g$^{-1}$, which was over two times larger than the value of uncoated BC particles suggested by Bond and Bergstrom, (2006) ($MAC_{BC}(880)_{uncoated}$ = 4.7 m$^2$ g$^{-1}$, extrapolated to 880 nm by assuming $AAE_{BC}$ = 1.0). This indicates that biomass-burning-related BC particles experienced substantial aging processes during their transport. The enhancement in $MAC_{BC}(880)_{biomass}$ can be explained by the 'lensing effect' due to the BC particles internally-mixed with other substances (Lack and Cappa, 2010).

The second source factor was characterized by large loadings of benzothiazolone (54%), Pb (46%), Br (40%), Cu (35%), Zn (27%), EC (36%), and OC (30%). The benzothiazolone is a substance released from the breaking-down of the antioxidant in vehicular tire (Cheng et al., 2006). Br is a tracer closely related



to motor vehicle emission (Guo et al., 2009), and Zn and Cu are associated with the combustion of lubricating and wearing of brake and tires (Lough et al., 2005; Song et al., 2006). Meanwhile, EC and OC also can be used to denote motor vehicle emissions (Cao et al., 2013). Although unleaded gasoline has been extensively used in China since 2005, a considerable portion of Pb is still found in vehicle-related

particles, which may be relation with the wear of metal alloys (Hao et al., 2019). Therefore, this source factor was identified as traffic emissions. This source constitutes a moderate percentage of primary $b_{abs}(\lambda)$ (15–30%). The estimated traffic emission-related AAE (AAE$_{traffic}$) was 0.8, consistent with the feature that BC is the dominant light-absorbing carbon for traffic emissions (Kirchstetter et al., 2004). The AAE$_{traffic}$ here is also close to those obtained with $\Delta^{14}C$ approach (Zotter et al., 2017). The estimated BC

MAC(880) of traffic emissions (MAC$_{BC}$(880)$_{traffic}$ = 9.1 m$^2$ g$^{-1}$) was similar with MAC$_{BC}$(880)$_{biomass}$, indicating that traffic emission-related BC particles were also subjected to substantial aging.

The third factor was dominated by high loadings of As (70%), S (37%), and Cu (47%), which is a typical feature of coal combustion (Hsu et al., 2016; Kim and Hopke, 2008). Although coal is a scarce energy used on the TP, it can be influenced by coal combustion from surrounding areas (e.g., East Asia, Li et al.,

2016). This source contributed only 12–17% of primary $b_{abs}(\lambda)$, less than biomass burning and traffic emissions. The obtained AAE of coal combustion (AAE$_{coal}$ = 1.1) was similar to AAE$_{traffic}$, suggesting that BC was also a dominant light-absorbing carbon in coal combustion emissions. The AAE$_{coal}$ is close to the value of chunk coal combustion (1.3) but lower to briquettes coal (2.6) (Sun et al., 2017), which may reflect the coal type that affected BC particles at Gaomeigu. The estimated BC MAC(880) of coal

combustion (MAC$_{BC}$(880)$_{coal}$ = 15.5 m$^2$ g$^{-1}$) was larger than MAC$_{BC}$(880)$_{biomass}$ and MAC$_{BC}$(880)$_{traffic}$. Numerous studies have confirmed that aged BC could result in MAC increased by a factor of 1.5 – 3.5 relative to uncoated one (Chen et al., 2017; Ma et al., 2020). The enhance factor for MAC$_{BC}$(880)$_{coal}$ (3.3) falls near the upper limit of this range. This large enhancement should be related to the aging processes of BC particles during their long-range transport to Gaomeigu, although more work is need to explain in

the future.

The fourth source factor was enriched of high loadings of Ca (35%), Ti (66%), Mn (47%), and Fe (61%), consistent with the characteristics of crustal elements (Guo et al., 2009). Thus, it was assigned to soil dust. The light absorption of soil dust is mainly due to the presence of iron oxides and varies with the diversity



of iron contents (Alfaro, 2004; Valenzuela et al., 2015). The mineral dust here contributed minor to primary $b_{abs}(\lambda)$ (6–9%) due to the low content of iron oxides, which was consistent with the result obtained on the southeastern TP (Zhao et al., 2019). The estimated AAE of mineral dust ($AAE_{dust}$) was 1.5, which is within the range of 1.2–3.0 obtained from multi-non-oceanic sites (Dubovik et al., 2002).

## 3.2 BC source apportionment

As fossil fuel-related BC aerosol involves BC emitted from traffic and coal combustion, the $AAE_{fossil}$ (0.9) and $MAC_{BC}(880)_{fossil}$ (12.3 m$^2$ g$^{-1}$) were averaged by the values of $AAE_{traffic}$ + $AAE_{coal}$ and $MAC_{BC}(880)_{traffic}$ + $MAC_{BC}(880)_{coal}$, respectively. Based on the source-specific AAEs (i.e., $AAE_{fossil}$ and $AAE_{biomass}$) and BC $MAC_{BC}(880)$ (i.e., $MAC_{BC}(880)_{fossil}$ and $MAC_{BC}(880)_{biomass}$), the mass concentrations of $BC_{fossil}$ and $BC_{biomass}$ were then estimated using the improved 'aethalometer model' (Eqs. (1) – (7)). As shown in Fig. S3a, no correlation ($r = 0.01$, $p = 0.02$) was found between $BC_{biomass}$ and $BC_{fossil}$, implying that BC from these two sources can be separated reasonably by the improved aethalometer model. The biomass-burning tracer (levoglucosan) and traffic-related tracer (benzothiazolone) were used to further verify the results of BC source apportionment. As expected, $BC_{biomass}$ and $BC_{fossil}$ correlated significantly with levoglucosan ($r = 0.75$, $p < 0.01$, Fig. S3b) and benzothiazolone ($r = 0.67$, $p < 0.01$, Fig. S3c), respectively. These results indicate that the source-specific AAEs and MACs(880) obtained from optical source apportionment were appropriate for the improved 'aethalometer model'.

Fig. 2 shows the time series of hourly averaged mass concentrations of BC, $BC_{biomass}$, and $BC_{fossil}$ during the campaign. The hourly BC mass concentration varied by almost 50-fold from 0.1 to 4.9 μg m$^{-3}$, with an arithmetic mean (± standard deviation) of 0.7 (± 0.5) μg m$^{-3}$, which was lower than those in the western TP but higher than in the northern TP (Wang et al., 2018, and references therein). The larger BC loading in the western TP is mainly attributed to the influences of Southeast Asia, where anthropogenic actives are intensive (Kurokawa et al., 2013). In terms of different BC sources, the mass concentrations of $BC_{biomass}$ (0.4 ± 0.3 μg m$^{-3}$, 57% of total BC) was slightly higher than $BC_{fossil}$ (0.3 ± 0.2 μg m$^{-3}$, 43% of total BC) on average. The mass fraction of $BC_{biomass}$ enhanced as the BC loading increased while $BC_{fossil}$ mass fraction showed an inverse trend (Fig. S4). This suggests and biomass-burning emissions were responsible for the high BC loading episode during the campaign.



Distinct diurnal variations in mass concentrations of $BC_{biomass}$ and $BC_{fossil}$ were observed (Fig. 3a). For $BC_{biomass}$, it started to increase after midnight reaching a small peak at 05:00 and then remained at a constant level before sunrise (~08:00). This may be attributed to the accumulative effect caused by planetary boundary layer (PBL) height (https://rda.ucar.edu/datasets/ds083.2), which showed a declining trend during this period (Fig. 3b). Thereafter, the $BC_{biomass}$ increased again and reached the maximum value of the day at midday. This enhancement was accompanied by the increased PBL height and wind speed (https://rda.ucar.edu/datasets/ds083.2) (Fig. 3b). Generally, deeper PBLs and stronger winds accelerate local pollutants diffusion and as a result lower their loadings (Wang et al., 2015). However, the increased trend here was more likely influenced by regional transport of $BC_{biomass}$ from upwind region. After sunrise, the PBL height began to rise and accompanied with west/southwest winds from 08:00 to 12:00 (Fig. S5). These favourable meteorological conditions provided good transport advantages for pollutants (including BC) from high-density biomass-burning emission areas to the sampling site (Chan et al., 2017). After the midday peak, $BC_{biomass}$ decreased sharply until midnight. The initial decrease (13:00 – 18:00) was mainly attributed to the continuous increases in PBL height and wind speed, which promoted $BC_{biomass}$ diffusion. The further reduction at night was due to the scarcity of local biomass-burning activities, although the PBL height and wind speed decreased during this period.

As shown in Fig. 3a, different from the unimodal diurnal pattern of $BC_{biomass}$, the $BC_{fossil}$ showed a roughly opposite trend. Owing to the influence of diffusion effect caused by increases in PBL height and wind speed, $BC_{fossil}$ decreased from 09:00 to 15:00. The initial morning decrease in $BC_{fossil}$ was just opposite to the increasing trend of $BC_{biomass}$, reflecting minor regional transport contributing to $BC_{fossil}$. Further, because of the small contribution of coal combustion to EC (12%, Fig.1c), the $BC_{fossil}$ was mainly contributed by motor vehicle emissions from the surrounding areas of southeastern TP. The subsequent increase in $BC_{fossil}$ from 17:00 to 20:00 was attributed to the reduction of PBL height and as a result accumulated pollutant in the southeastern TP. Since there were scarce traffic activities at night, the $BC_{fossil}$ remained at a steady concentration from 21:00 to 08:00. The stable nocturnal $BC_{fossil}$ may reflect the impact of fossil fuel emissions on BC in the southeastern margin of TP due to the accumulation effect driven by the low PBL height.



### 3.3 Regional influences of BC$_{biomass}$ and BC$_{fossil}$

To investigate the regional impacts on BC, three groups of air masses were identified based on their transport pathways (Fig. 4a). Cluster #1 originated from northeastern India and then passed over Bangladesh before arriving Gaomeigu. The average BC mass concentration of this cluster was the highest (0.8 ± 0.4 μg m$^{-3}$) among the three clusters. About 74% of total trajectories were associated with Cluster #1, of which 22% was identified as polluted ones with an average BC loading of 1.3 ± 0.5 μg m$^{-3}$. Cluster #2 derived from Burma with an average BC loading of 0.7 ± 0.7 μg m$^{-3}$. This cluster constituted only 24% of total trajectories, but among them, about 37% was referred to pollution with BC reaching as high as 1.6 ± 0.9 μg m$^{-3}$. The air masses associated with Cluster #3 originated from interior of mainland China, which had the lowest BC mass concentration of 0.4 ± 0.1 μg m$^{-3}$. This cluster comprised small fraction of total trajectories (2%), and none of them was identified as pollution, suggesting minor influence from the mainland China during the campaign.

The diurnal patterns of BC$_{biomass}$ and BC$_{fossil}$ from different clusters were further used to investigate the impacts of regional transport. As shown in Fig. 3c and e, a similar diurnal variation of BC$_{biomass}$ was found for Cluster #1 and Cluster #2, with larger values during daytime compared with the night-time. The enhancements of daytime BC$_{biomass}$ (8:00 – 12:00) were related to the regional transport from northeastern India (Cluster #1) and Burma (Cluster #2). For Cluster #3, the diurnal variation of BC$_{biomass}$ decreased during the daytime and increased at night (Fig. 3g), which was mainly controlled by the daily variability in PBL height. Compared to other two clusters, the inverse diurnal variation of BC$_{biomass}$ for Cluster #3 indicates the influences of biomass-burning activities from the surrounding areas rather than regional transport. However, it should be noted that these cases were scarce because of only 2% of air-masses associated with Cluster #3. For BC$_{fossil}$, a similar diurnal pattern was found for Cluster #1 and Cluster #2 (Fig. 3c and e), which was mainly associated with the influences of traffic emissions in surrounding areas as well as daily cycles of PBL as discussed in section 3.2. The BC$_{fossil}$ of Cluster #3 (Fig.3g) exhibited a relatively stable diurnal variation expect for sporadic fluctuations. Unlike the declining trend of BC$_{fossil}$ during the daytime in the other two clusters, the relative stable variation in Cluster #3 indicates that there were fossil fuel sources to offset the effect of pollutant diffusion caused by increased PBL height. Owing





to the high-density transportation network in mainland China (Liu, 2019), the regional transport of traffic emissions may be the cause.

Further, the PSCF model was applied to investigate the likely spatial distribution of pollution source regions for $BC_{biomass}$ and $BC_{fossil}$. As shown in Fig. 4b, a low PSCF value of $BC_{biomass}$ was found near Gaomeigu while high values were concentrated in the northeastern India and northern Burma, consistent with intensive fire spots in these areas (Fig. S6). This indicates that large $BC_{biomass}$ loadings at Gaomeigu were more likely influenced by cross-border transport of biomass burning rather than local emissions. For $BC_{fossil}$ (Fig. 4c), the most possible impact region was located near the southeast of Gaomeigu, where there are a few villages scattered around and two highways (e.g., Hangzhou-Ruili Expressway and Dali-Nujiang Expressway). Owing to the low consumption of coal in the southeastern TP (Li et al., 2016), the high PSCF values of $BC_{fossil}$ may be mainly contributed by traffic emissions rather than coal combustion. Moreover, sporadic high PSCF values of $BC_{fossil}$ were also found in the northern Burma, indicating possible influences of fossil fuel emissions.

To further quantify the contributions of Southeast Asia transported to BC at Gaomeigu, a high BC episode (23 – 27 March) was arbitrary selected and simulated by the WRF-Chem model. Two scenarios of emissions were simulated: one involved all BC emission sources, and the other turned off biomass-burning emissions in Southeast Asia. The variation of modelled BC mass concentration shows an acceptable consistency with the measured one ($r = 0.63$, $p < 0.01$, Fig. S7), and the index of agreement was estimated to be 0.77, indicating that the formation process of this BC episode was captured by the WRF-Chem model. However, it should be noted that the normalized mean bias between the measured and modelled BC values was estimated to be 24%, suggesting an overestimation of simulation. This discrepancy could be attributed to the simulation uncertainties resulted from the emission inventory and meteorological conditions. Fig. 5a shows the spatial distributions of BC loadings in Gaomeigu and surrounding areas. The mass concentrations of BC can exceed 15 μg m$^{-3}$ in Burma and northern India, which was over one order of magnitude higher than that in the southeastern margin of TP (0.7 μg m$^{-3}$). After turning off the biomass burning emission source in Southeast Asia, the BC loading at southeastern TP dropped over 40% (Fig. 5b), suggesting a substantial impact of biomass-burning activities in Southeast Asia countries on BC at the sampling site, which was consistent with cluster analysis and PSCF results.





### 3.4 Radiative forcing and heating rate

Fig. 6 shows the average DRFs of PM$_{2.5}$ and BC (including BC$_{biomass}$ and BC$_{fossil}$) at the TOA and SUF during the campaign. The average DRF of PM$_{2.5}$ at the TOA was +0.03 ± 1.1 W m$^{-2}$, implying the importance of light-absorbing carbon. Actually, the BC can produce +1.6 ± 0.8 W m$^{-2}$ at the TOA on

average. At the SUF, BC forcing (-3.0 ± 1.5 W m$^{-2}$) contributes near half of PM$_{2.5}$ forcing (-6.3 ± 4.5 W m$^{-2}$). The net forcing trapped in the atmosphere was +4.6 ± 2.4 W m$^{-2}$ for BC aerosol, which comprised 73% of that caused by PM$_{2.5}$ (6.3 ± 4 W m$^{-2}$), suggesting a substantial radiative effect caused by BC in the southeastern margin of TP, although its mass fraction is small in PM$_{2.5}$ (3.3%).

From the perspective of BC sources (Fig. 6), the average BC$_{biomass}$ (BC$_{fossil}$) DRF was +0.8 ± 0.6 W m$^{-2}$

(+0.7 ± 0.4 W m$^{-2}$) at the TOA, and -1.7 ± 1.2 W m$^{-2}$ (-1.4 ± 0.6 W m$^{-2}$) at the SUF. This causes an average atmospheric radiative forcing of +2.5 ± 1.8 W m$^{-2}$ (+2.1 ± 0.9 W m$^{-2}$). Owing to the impacts of regional transport, the atmospheric forcing of BC$_{biomass}$ was more fluctuation relative to BC$_{fossil}$ (Fig. S8). For example, the average atmospheric forcing of BC$_{biomass}$ can reach as high as +5.6 W m$^{-2}$ when the coming air masses passed over the biomass-burning regions in Southeast Asia.

The atmospheric forcing indicates the trapping of energy in the atmosphere leading to an increase in atmospheric heating in the southeastern margin of TP. The heating rate caused by BC varied from 0.02 to 0.3 K day$^{-1}$ with an average of 0.13 ± 0.07 K day$^{-1}$. In terms of the DRF efficiency, the heating rate caused by a unit BC mass concentration in this region is found higher (0.19 (K day$^{-1}$) (μg m$^{-3}$)$^{-1}$) compared with that in Qinghai Lake, northeastern TP (0.13 (K day$^{-1}$) (μg m$^{-3}$)$^{-1}$) (Wang et al., 2015), but it was

generally lower than the values in the southwest of Himalaya regions (Fig. S9). The heating rate caused by the BC$_{biomass}$ is a slightly higher (0.07 ± 0.05 K day$^{-1}$) compared with BC$_{fossil}$ (0.06 ± 0.02 K day$^{-1}$). The heating rate of BC$_{biomass}$ increased to 0.16 K day$^{-1}$ when the BC mass concentration was heavily influenced by the polluted air mass from Southeast Asia.

### 4 Conclusions

This study explored the source contribution and the direct radiative forcing of the black carbon (BC) aerosol from fossil fuel combustion and biomass burning in the southeastern margin of Tibetan Plateau. The observed mean BC concentration was 0.7 ± 0.5 μg m$^{-3}$ during the campaign. Based on the optical





source apportionment using multi-wavelength absorption and chemical species, the obtained absorption Ångström exponents (AAEs) and BC mass absorption cross section (MAC) at wavelength of 880 nm were 0.9 and 12.3 $m^2\ g^{-1}$ for fossil fuel source and 1.7 and 10.4 $m^2\ g^{-1}$ for biomass burning, respectively. Using these source-specific AAEs and BC MACs, the improved aethalometer model estimated 43% of

BC from fossil fuel source and 57% from biomass burning ($BC_{fossil}$ and $BC_{biomass}$, respectively). The diurnal cycle in $BC_{biomass}$ was driven by BC regional transport and the variation of planetary boundary layer (PBL) height. In contrast, $BC_{fossil}$ showed a roughly reverse diurnal cycle relative to $BC_{biomass}$, which was mainly due to traffic emissions in surrounding areas and the variation of PBL height.

Three groups of air masses were identified based on their transport pathways, with the highest BC loading

from the direction of northeastern India, followed by Burma and with the lowest from mainland China. The potential sources contribution function (PSCF) model showed that the northeastern India and northern Burma were the most possible contribution areas to $BC_{biomass}$ at the sampling site, and the Weather Research and Forecasting model coupled with chemistry (WRF-Chem) model indicates that biomass burning from the Southeast Asia can contribute 40% of BC loading at the sampling site. The

southeast of Gaomeigu was the most likely contribution region for the high $BC_{fossil}$.

The Santa Barbara DISORT Atmospheric Radiative Transfer (SBDART) model estimated a cooling effect of $-3.0 \pm 1.5$ W $m^{-2}$ at the Earth's surface and a warming effect of $+1.6 \pm 0.8$ W $m^{-2}$ at the top of the atmosphere caused by BC aerosol. The atmospheric forcing of BC was $+4.6 \pm 2.4$ W $m^{-2}$, comprising 73% of that caused by $PM_{2.5}$. The average $BC_{biomass}$ DRF was higher than that caused by $BC_{fossil}$. The

average heating rate caused by BC was $0.13 \pm 0.07$ K $day^{-1}$, with $0.07 \pm 0.05$ K $day^{-1}$ from $BC_{biomass}$ and $0.06 \pm 0.02$ K $day^{-1}$ from $BC_{fossil}$. The heating rate of $BC_{biomass}$ increased can increase to 0.16 K $day^{-1}$, when BC mass concentration was heavily influenced by the emission of biomass burning from Southeast Asia.

*Data availability*. All data described in this study are available upon request from the corresponding authors.

*Supplement*. The supplement related to this article is available online.



*Author contributions*. QW and JC designed the study. WR conducted the field measurements. LX provided the results of WRF-Chem model. YZ and TZ performed the chemical analysis of filters. HL and QW wrote the article. All the authors reviewed and commented on the paper.

*Competing interests*. The authors declare that they have no conflict of interest.

*Acknowledgments*. This work was supported by the National Natural Science Foundation of China (41877391), the Second Tibetan Plateau Scientific Expedition and Research Program (STEP) (2019QZKK0602), and the Youth Innovation Promotion Association of the Chinese Academy of Sciences (2019402).

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



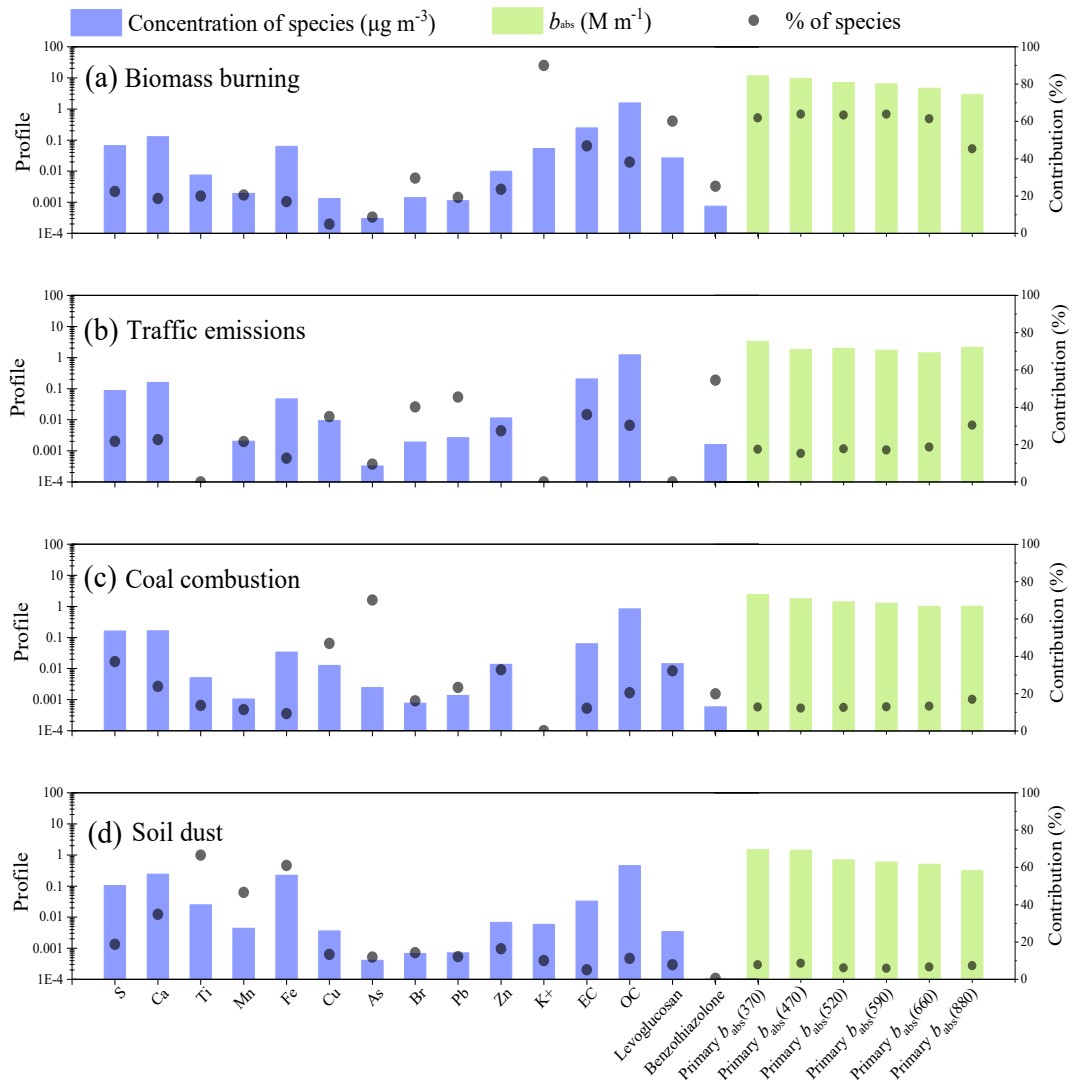

**Figure 1.** Four sources identified by a positive matrix factorization (PMF). Concentration ($\mu g\ m^{-3}$) of the chemical species in each source are colored by purple. Primary $b_{abs}$ ($\lambda$) at six wavelengths ($\lambda = 370$, 470, 520, 590, 660, or 880nm) in each source ($M\ m^{-1}$) are colored by green.





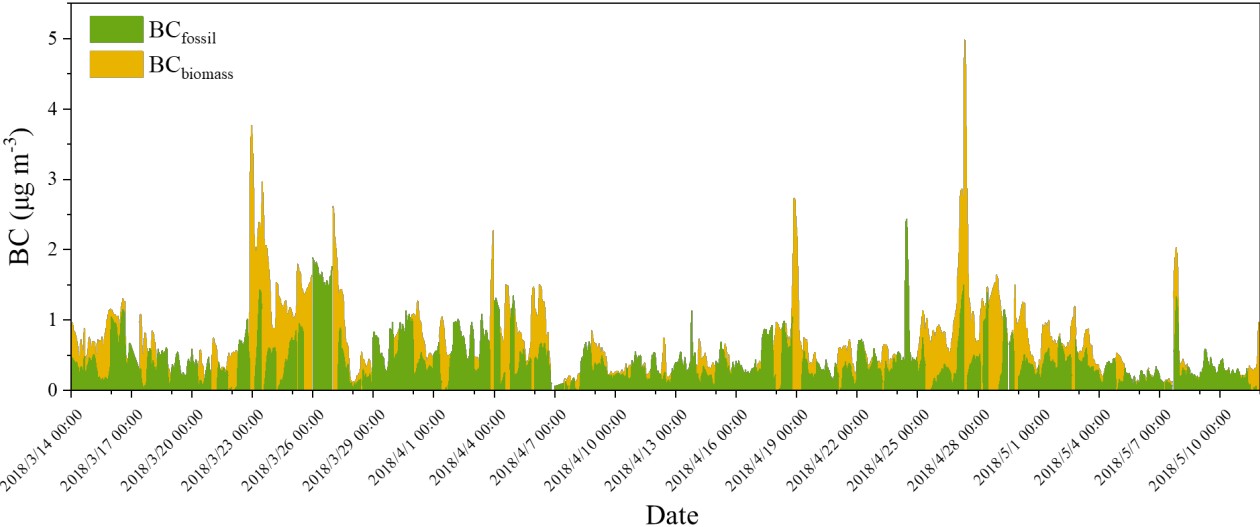

**Figure 2.** Time series of hourly averaged mass concentrations of black carbon (BC) aerosol from biomass bunirng ($BC_{biomass}$) and fossil fuel sources ($BC_{fossil}$) during the campaign.

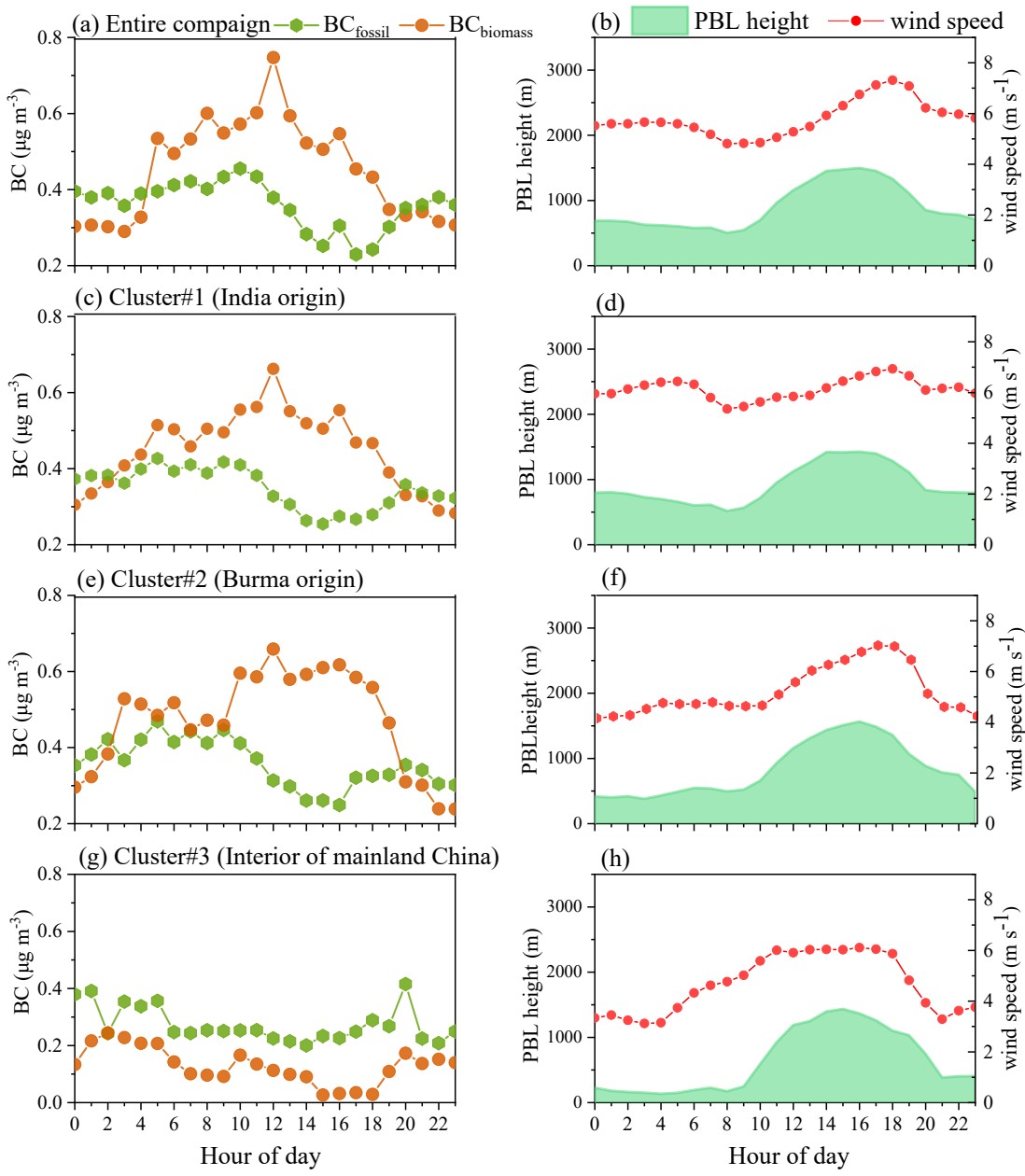

**Figure 3.** (left panel) Diurnal variations of hourly averaged black carbon (BC) aerosol from biomass burning (BC$_{biomass}$) and fossil fuel sources (BC$_{fossil}$) as well as (right panel) wind speed and planetary boundary layer (PBL) height during the entire campaign and different air-mass directions.





**Figure 4.** Maps of (a) the mean trajectory clusters and the potential source contribution function for black
carbon (BC) aerosol from (b) biomass burning (BC$_{biomass}$) and (c) fossil fuel sources (BC$_{fossil}$) during the
campaign.



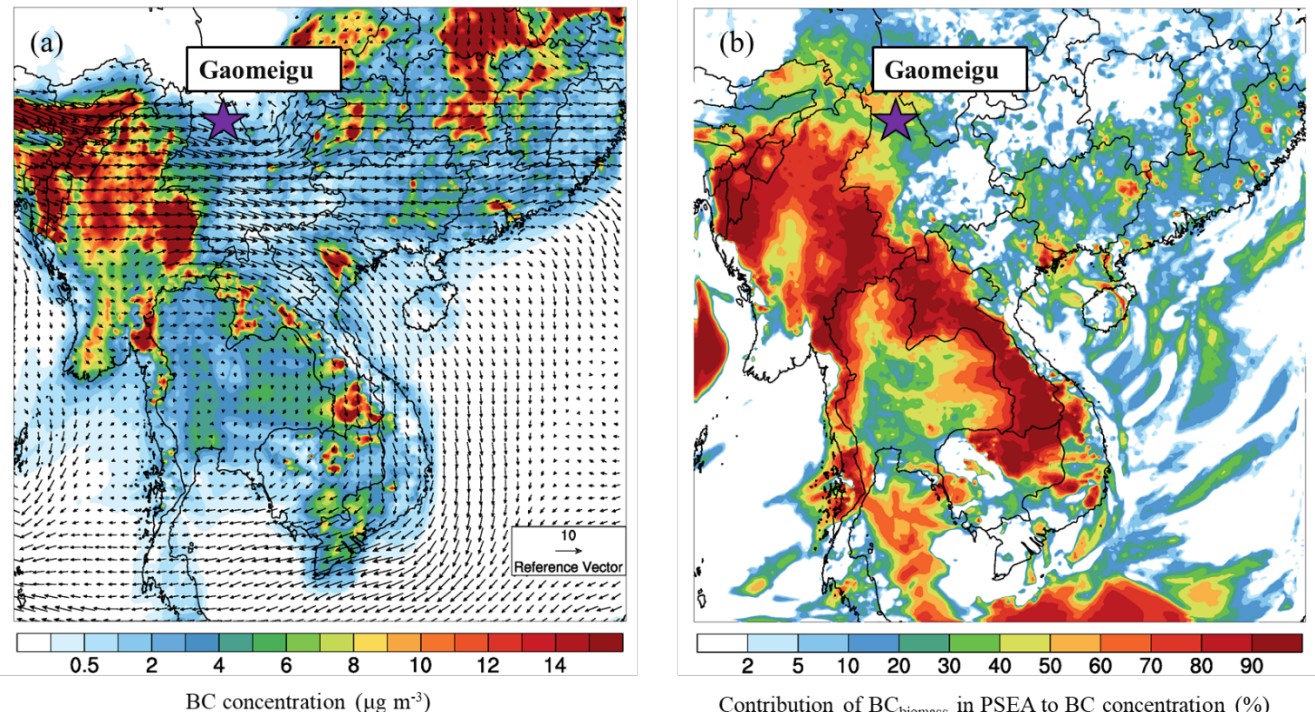

**Figure 5.** (a) Spatial distributions of simulated black carbon (BC) mass concentrations in Southeast Asia and (b) the mass contribution of biomass-burning BC ($BC_{biomass}$) in peninsular Southeast Asia (PSEA). The simulated surface winds were overlaid on (a).



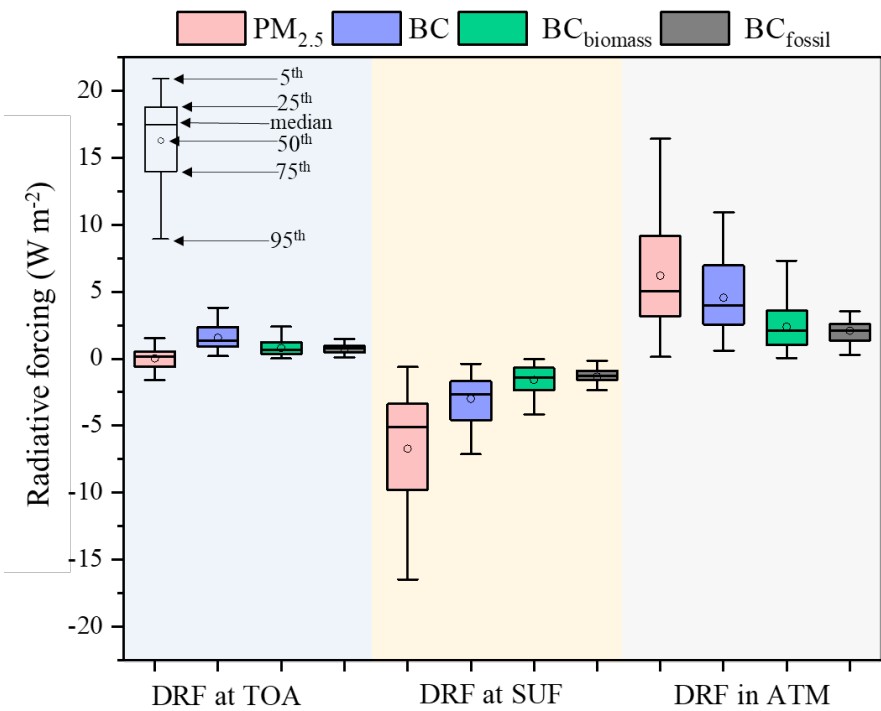

**Figure 6.** The average direct radiative forcing (DRF) of black carbon (BC) aerosol and PM$_{2.5}$ at the Earth's surface (SUF), the top of atmsphere (TOA), and in the atmosphere (ATM). The BC$_{biomass}$ and BC$_{fossil}$ represent BC contributed by biomass burning and fossil fuel sources, respectively.