# Peer review of "Measurement report: Quantifying source contribution and radiative forcing of fossil fuel and biomass burning black carbon aerosol in the southeastern margin of Tibetan Plateau"

_Atmospheric Chemistry and Physics, 2020_

## Referee Comment (RC1) · Anonymous Referee #1 · 15 Jul 2020

General comments: In this manuscript, the sources of BC aerosols over the Tibetan Plateau and their radiative effects were investigated. BC aerosols were distinguished into fossil fuel combustion source and biomass burning source. Regional transport of source-specific BC was further explored by models. On this basis, the radiative effects caused by BC from different sources were evaluated. Overall, the manuscript is well structured, the methods are technically sound, and the main findings presented seem to be reasonable and be of general interests to the Tibetan Plateau ecosystem and climate stability study. I think the topic fits within the scope of ACP. I would

recommend acceptance of this manuscript for publication pending the following revisions: Specific comments: 1.Please spend time picking through the manuscript and check for spelling and grammatical errors, especially the tense, prepositions and articles. For example, 'BC on the TP' should be replaced with 'BC over the TP', 'transport to TP' should be replaced with 'transport to the TP'... 2.Section 1, this part should introduce the research background and significance, current status, concealed problems, as well as research mentality and content of this study. Please highlight the innovation and importance from another angle and reduce describing the deficiencies of previous research appropriately. 3.Please try to avoid expressions like 'our study', which seems not be objective. Technical corrections: 1.P1, Line 24, 'reveal' should be changed to 'revealed'. 2.P1, Line 26, add 'which' before 'can explain'. 3.P2, Line 1, delete 'and' before 'heating rates of'. 4.P2, Line 1, 'The glaciers on the TP recently shows are rapidly retreating' should be revised. 5.P2, Line 4-8, Please cite these literatures, doi:10.1093/nsr/nwz191, doi.org/10.1016/j.atmosenv.2020.117583, doi: 10.1016/j.atmosenv.2019.04.001. 6.P2, Line 8-10, Please cite these literatures, doi.org/10.3390/rs12020231, doi:10.1002/joc.6430. 7.P2, Line 16, 'the atmospheric BC studies on the TP' should be changed to 'studies on the TP atmospheric BC'. 8.P2, Line 16-18, Please cite the literature, doi: 10.5194/acp-15-12581-2015. 9.P2, Line 24, 'the other is based on the field observations to apportion BC into different sources through a certain data analytical method.' Please add appropriate literature. 10.P2, Line 25, 'transport to TP' should be changed to 'transport to the TP'. 11.P3, Line 5, 'are advantageous to capturing' should be changed to 'are advantageous to capture'. 12.P3, Line 10, 'cross-section (MAC) used in the model.' Please add appropriate literature. 13. P4, Line 11, 'on the rooftop of' should be changed to 'at on the rooftop of'. 14. P4, Line 15, 'the radiative effect' should be changed to 'the radiative effects'. 15. P5, Line 4, 'was resolved using' should be changed to 'was resolved by using'. 16. P8, Line 10, 'the number of the endpoints' should be changed to 'the number of endpoints'. 17. P9, Line 8, 'model is elaborated in Ricchiazzi and Yang, (1998)' should be changed to 'model was elaborated in Ricchiazzi and Yang (1998)'. 18. P10, Line 15, delete 'which

is' before 'within a relative boarder range'. P10, Line 18, delete 'which was' before 'over two times'. In addition, it is necessary to pay attention to the tense errors, which often appear throughout the manuscript. 19. P10, Line 19, Line 85, the comma before '(2006)' should be deleted. 20. P11, Line 5, 'may be relation with' should be changed to 'may be related with' or 'may have relation with'. 21. P12, Line 26, 'This suggests and' should be changed to 'This suggests that'. 22. P14, Line 12, 'the mainland China' should be changed to 'mainland China' or 'the mainland of China'.

---

## Referee Comment (RC2) · Anonymous Referee #2 · 11 Aug 2020

Review of "Measurement report: Quantifying source contribution and radiative forcing of fossil fuel and biomass burning black carbon aerosol in the southeastern margin of Tibetan Plateau" (ACPD-2020-408, Liu et al) General comment This paper reports on measurements and modeling regarding the contribution of fossil fuel and biomass burning sources to black carbon (BC) aerosols abundance and radiative forcing at a site south-east of the Tibetan Plateau. Methods used in the study are robust, and the results are sound. However, it is difficult to ascertain the novelty and actual contribution to the overall understanding of, for instance, how BC aerosols are affecting

the Tibetan Plateau. In my opinion, authors may turn this study into a relevant one if they would consider using the data at hand by better explaining the reasons that make these data important for improved understanding. In its present form, detailed measurements and modeling are more suited for a technical report not suitable, in my opinion, for this prestigious journal. Specific comments • The text would improve in clarity and possibly be shorten if reviewed by a native English writer/speaker. Also, some results could be summarized in tables improving the readability of the text. • Abstract: Re-write according to suggestions below. What do we learn from this study? In what context is this useful? What is the novelty? • Introduction o Page 2, lines 13-14. In addition to characterizing source regions and their contributions to aerosol burden downwind it is also important to assess the timing in which this impact occur, how is the aging process, etc. o Page 2, lines 22-24. Uncertainties in modeling studies not only depend on uncertain emission estimates but also on how well chemistry, transport and deposition processes are represented, initial/boundary conditions, etc. It appears necessary to review other studies to get an idea of the uncertainty when using models to simulate long-range transport, particularly over complex terrain. o Page 3, lines 1-24. This is a lengthy discussion about distinguishing between biomass and fossil fuel black carbon according to multiple observational and methodological studies. Rather than listing the pros and cons of the different methods, it would be good to have a clearer idea of which is the method fit for purpose to be discussed in the work. For that, it is key to establish a clear purpose, and how this will help improving understanding of a given phenomenon. o Page 4, lines 1-6. You state that previous studies have dealt with radiative impacts of bulk BC, no distinguishing BC sources. Furthermore, you state that this study would be unique as it provides the first estimate of BC radiative forcing split by source regions. However, you estimate the instantaneous forcing over one site which is, by definition, locally representative, and not necessarily climatically important. Other studies may have estimated bulk BC forcing but over much longer periods of time, and over large areas, including the Himalayan cryosphere. Hence, I urge the authors to make their study unique by

better establishing the purpose of it. • Methodology o Page 4, line 17. Improve the precision of attitude and longitude to allow a proper location of the site. o Page 4, lines 18-19. As per your reference, Wang et al (2019a), your observation site is located along a "transportation channel". Describe the overall transport patterns affect. Is the period of observations representative of which transport/circulation pattern? An overall meteorological description is missing. o Page 4, lines 19-20. You say that the population surrounding your observational site is small. Small compared to what? Then you go onto establishing that limited anthropogenic activities are found there. However, your results show a non-negligible contribution. The site should be better described, including a brief description of aerosol sources. o Page 7, section 2.5. HYSPLIT can be used with large-scale (synoptic) meteorological fields. Do you have an assessment of how well this approach works over complex terrain? Why do you choose 3-day back trajectories instead of 2 or 5 days? • Results and discussion o Page 11, lines 20-25. Your BC aerosol appears to have aged. Can't you use your WRF-Chem simulations to attempt providing further insights on this issue? o Page 12, section 3.2.  Some of your results could be better appreciated if summarized in a table.  You make multiple references to FigS3. Maybe it is better to bring it to the main manuscript. If so, it could be useful to split the graphs for daytime and nighttime periods as it would better fit with Figure 2. • Conclusion o Stress the novelty, and make it explicit that the period studied correspond to a given set of transport/circulation patterns.

Please also note the supplement to this comment:
https://acp.copernicus.org/preprints/acp-2020-408/acp-2020-408-RC2-supplement.pdf

---

## Author Comment (AC1) · 29 Oct 2020

General comments: In this manuscript, the sources of BC aerosols over the Tibetan Plateau and their radiative effects were investigated. BC aerosols were distinguished into fossil fuel combustion source and biomass burning source. Regional transport of source-specific BC was further explored by models. On this basis, the radiative effects caused by BC from different sources were evaluated. Overall, the manuscript is well structured, the methods are technically sound, and the main findings presented seem to be reasonable and be of general interests to the Tibetan Plateau ecosystem and climate stability study. I think the topic fits within the scope of ACP. I would recommend acceptance of this manuscript for publication pending the following revisions:

**Response:** We sincerely thank the reviewer for the comments and suggestions, and we have revised the relevant text and content. Below are point-to-point responses, and the modifications to the manuscript are marked.

**Specific comments:**

1.Please spend time picking through the manuscript and check for spelling and grammatical errors, especially the tense, prepositions and articles. For example, 'BC on the TP' should be replaced with 'BC over the TP', 'transport to TP' should be replaced with 'transport to the TP'

**Response:** We have taken the suggestion to heart and have corrected the relevant mistakes as shown below. Also, the paper has been polished by a native English speaker.

"Therefore, quantitative information on the contributions of different sources of BC over the TP is lacking, but it is critically needed for a better understanding the influence of anthropogenic emissions on its environment and climate."

"Nonetheless, these studies have been helpful for understanding the sources of BC over the TP."

2.Section 1, this part should introduce the research background and significance, current status, concealed problems, as well as research mentality and content of this study. Please highlight the innovation and importance from another angle and reduce describing the deficiencies of previous research appropriately.

**Response:** We have re-written the introduction to include more background material, to clarify the research focus, to explain why we were interested in study area and why we chose the methods we did. The revised introduction now reads:

"The Tibetan Plateau (TP) is an important regulator of climate change in the northern hemisphere, and it plays crucial roles in the functions of global ecosystems and climate stability (Liu et al., 2019b). The TP is covered by one of the largest ice masses on Earth, and it has been called the water tower of Asia (Liu et al., 2020). The glaciers on the TP are facing rapid retreat, however, and if unchecked, that could result in adverse effects on Asian hydrological cycle and the Asian monsoon (Luo et al., 2020). In spring, the glaciers on the TP begin to melt as part of the natural hydrological cycle, but the increasing quantities of black carbon (BC) aerosol transported to the TP has accelerated this process (Bond et al., 2013) by causing a warming effect in atmosphere over the TP and enhancing the absorption of radiation on the surface of the glaciers (Ming et al., 2009).

The southern part of the TP is bounded by South Asia where air pollution often is severe (Chan et al. 2017). Several studies have shown that pollutants (including BC) from South Asia can be transported to the south of the TP along mountain-valleys, especially during the pre-monsoon (March-May) when southwestly winds prevail (e.g., Cao et al., 2010; Xia et al., 2011 Zhu et al., 2017; Niu et al. 2017). For example, Xia et al., (2011) analyzed satellite data and air mass trajectories and found that the TP, particularly the southern TP, was affected by pollutants carried by southwesterly winds from nearby regions in South Asia. In addition, numerous studies have shown that the high bulk BC mass loadings and the associated regional influences on the TP are related to transport from South Asia (Liu et al., 2015; Han et al., 2020; Cong et al., 2015; Wang et al., 2015). Nonetheless, assessments of regional transport of bulk BC aerosol have not fully revealed the impacts of different BC emission sources because the optical properties and radiative effects of BC not only can vary among sources in complex ways but also can be affected by aging during transport (Tian et al., 2019; Zhang et al., 2019). Therefore, quantitative information on the contributions of different sources of BC over the TP is lacking, but it is critically needed for a better understanding the influence of anthropogenic emissions on its environment and climate.

Several studies have assessed the contributions of different BC sources through model simulations or isotopic methods. For example, Zhang et al. (2015) investigated BC sources for different parts of the TP by using a chemical transport model and a source tagging approach, and they found that the contributions of BC sources varied among regions and with the seasons. Li et al., (2016) used filter sampling and carbon isotopes ( $\Delta^{14}$ C and  $\delta^{13}$ C) to determine the BC from fossil

fuels and biomass burning in several areas of the TP. A major disadvantage of filterbased measurements is they are constrained by low time resolution, and this makes it challenging to capture the detailed evolution of pollution events. On the other hand, the accuracy of model simulations is dependent on many factors, including uncertainties associated with initial particle parameters, aging processes, the accuracy of emission inventory, meteorological fields over the complex terrain, and the modules for chemistry and planetary boundary layer (PBL) dynamics, etc. (Koch et al., 2009; Madala et al., 2014; Vignati et al., 2010). Nonetheless, these studies have been helpful for understanding the sources of BC over the TP.

To make up for the deficiencies of filter-based analysis, BC source apportionments based on high-time resolution online data has been conducted in many locations but for the TP are limited. An 'aethalometer model' based on multi-wavelength absorption data is one of efficient approaches for distinguishing between BC from fossil fuel and biomass burning sources (Sandradewi et. al., 2008). Although it has been widely used elsewhere, this approach has not been applied to the TP. The accuracy of the 'aethalometer model' relies on the input parameters, including the Ångström exponents (AAE) and BC mass absorption cross-sections (MACBC) of different sources (Zotter et al., 2017). Limited information on site specific AAEs and MACBCS, lead most studies to rely on values taken from measurements made in other locations (e.g., Healy et al., 2017; Zhu et al., 2017). This results in unquantified uncertainties because the AAEs and MACBCS vary with specific fuel subtypes and combustion conditions (Wang et al., 2018; Tian et al., 2019). Therefore, site-dependent AAE and MACBC are essential for improving the reliability of BC source apportionment by the 'aethalometer model'.

In this study, field measurements of BC were taken on the southeastern margin of the TP during the pre-monsoon. This region connects the high altitude TP with the low altitude Yungui Plateau and forms a transport channel for pollutants from Southeast Asia (Wang et al., 2019a), and it is an ideal region for investigating the impact of pollutant transport to the southeastern TP. A receptor model that combined multi-wavelength absorption with aerosol species concentrations was used to retrieve site-dependent AAEs and MACBCs. This was done to improve the 'aethalometer model' with the goal of obtaining a more accurate BC source apportionment. The primary objectives of this study were to (1) quantify the mass concentrations of BC from fossil fuel and biomass burning sources; (2) determine the impact of regional transport on source-specific BC; and (3) assess the radiative effects caused by BC from different sources. This study provides insights into the BC sources on southeastern TP and an assessment of their radiative effects during the pre-monsoon."

3.Please try to avoid expressions like 'our study', which seems not be objective. Technical corrections: 1.P1, Line 24, 'reveal' should be changed to 'revealed'.

**Response:** We have changed all "our study" in the manuscript into "this study". The verbal tenses have been correct in the following sentence:

"The potential source contribution function indicated that BCbiomass was transported to the site from northeastern India and northern Burma."

2.P1, Line 26, add 'which' before 'can explain'.

**Response:** We have changed the sentence to

"The Weather Research and Forecasting model coupled with chemistry (WRF-Chem) model indicated that 40% of the BCbiomass originated from Southeast Asia"

3.P2, Line 1, delete 'and' before 'heating rates of'.

**Response:** We have changed this sentence to

"The DRE of BCbiomass and BCfossil produced heating rates of  $0.07 \pm 0.05$  and  $0.06 \pm 0.02$  K day-1, respectively."

4.P2, Line 1, 'The glaciers on the TP recently shows are rapidly retreating' should be revised.

**Response:** We have corrected this to

"The glaciers on the TP are facing rapid retreat, however, and if unchecked, that could result in adverse effects on Asian hydrological cycle and the Asian monsoon (Luo et al., 2020)."

5.P2, Line 4-8, Please cite these literatures, doi:10.1093/nsr/nwz191, doi.org/10.1016/j.atmosenv.2020.117583, doi: 10.1016/j.atmosenv.2019.04.001.

**Response:** We cite those papers in the revised version.

"The Tibetan Plateau (TP) is an important regulator of climate change in the northern hemisphere, and it plays crucial roles in the functions of global ecosystems and climate stability (Liu et al., 2019b). The TP is covered by one of the largest ice masses on Earth, and it has been called the water tower of Asia (Liu et al., 2020)."

6.P2, Line 8-10, Please cite these literatures, doi.org/10.3390/rs12020231, doi:10.1002/joc.6430.

**Response:** We now cited these references in the revised version.

"The glaciers on the TP are facing rapid retreat, however, and if unchecked, that could result in adverse effects on Asian hydrological cycle and the Asian monsoon (Luo et al., 2020)."

7.P2, Line 16, 'the atmospheric BC studies on the TP' should be changed to 'studies on the TP atmospheric BC'.

**Response:** We have revised the sentence

"In addition, numerous studies have shown that the high bulk BC mass loadings and the associated regional influences on the TP are related to transport from South Asia (Liu et al., 2015; Han et al., 2020; Cong et al., 2015; Wang et al., 2015)."

8.P2, Line 16-18, Please cite the literature, doi: 10.5194/acp-15-12581-2015.

**Response:** We have read that paper and now cite it. Please see it in the answer for the above question.

9.P2, Line 24, 'the other is based on the field observations to apportion BC into different sources through a certain data analytical method.' Please add appropriate literature.

**Response:** We have re-written this text, and now it reads as follows:

"Several studies have assessed the contributions of different BC sources through model simulations or isotopic methods. For example, Zhang et al. (2015) investigated BC sources for different parts of the TP by using a chemical transport model and a source tagging approach, and they found that the contributions of BC sources varied among regions and with the seasons. Li et al., (2016) used filter sampling and carbon isotopes ( $\Delta^{14}$ C and  $\delta^{13}$  C) to determine the BC from fossil fuels and biomass burning in several areas of the TP."

10.P2, Line 25, 'transport to TP' should be changed to 'transport to the TP'.

**Response:** We have corrected and add the article "the" into TP in the revised version.

11.P3, Line 5, 'are advantageous to capturing' should be changed to 'are advantageous to capture'.

**Response:** We revised the sentence

"A major disadvantage of filter-based measurements is they are constrained by low time resolution, and this makes it challenging to capture the detailed evolution of pollution events."

12.P3, Line 10, 'cross-section (MAC) used in the model.' Please add appropriate literature.

**Response:** We have added the relevant literature:

"The accuracy of the 'aethalometer model' relies on the input parameters, including the Ångström exponents (AAE) and BC mass absorption cross-sections (MACBC) of different sources (Zotter et al., 2017)."

13. P4, Line 11, 'on the rooftop of' should be changed to 'at on the rooftop of'.

Response: We have corrected "on" into "at". The sentence now reads

"Intensive field measurements were made at the rooftop of a building (~10 m above the ground) at the Lijiang Astronomical Station, Chinese Academy of Sciences (3260 m above sea level, 100°1'48"E, 26°41'24"N), Gaomeigu County, Yunnan Province, China (Fig. 1) from 14 March to 13 May 2018."

14. P4, Line 15, 'the radiative effect' should be changed to 'the radiative effects'.

**Response:** We have changed all "radiative effect" into "radiative effects" in the revised version.

15. P5, Line 4, 'was resolved using' should be changed to 'was resolved by using'.

**Response:** We have changed the sentence as below:

"A dual-spot technique for the aethalometer measurements was used to compensate for non-linearity, while a factor of 2.14 was used to correct for the artifacts caused by the quartz filters (Drinovec et al., 2015)."

16. P8, Line 10, 'the number of the endpoints' should be changed to 'the number of endpoints'.

**Response:** We have deleted "the" before endpoints in the sentence:

"where  $m_{ij}$  is the number of endpoints associated with BC mass concentration higher than the set criterion;"

17. P9, Line 8, 'model is elaborated in Ricchiazzi and Yang, (1998)' should be changed to 'model was elaborated in Ricchiazzi and Yang (1998)'.

**Response:** We have corrected the tense as follows:

"The direct radiative effects (DRE) of source-specific BC were estimated with the widely used Santa Barbara DISORT Atmospheric Radiative Transfer (SBDART) model—a detailed description of which may be found in Ricchiazzi and Yang (1998)."

18. P10, Line 15, delete 'which is' before 'within a relative boarder range'. P10, Line 18, delete 'which was' before 'over two times'. In addition, it is necessary to pay attention to the tense errors, which often appear throughout the manuscript.

**Response:** Thanks for pointing out the error in tense. The paper has been edited and thoroughly revised, and the problems with tense should have been corrected. The sentences were changed to

"That was within the relatively broad range of AAEbiomass (1.2–3.5) determined by other methods (e.g.,  $\Delta^{14}$ C and organic tracers) in previous studies (Sandradewi et al., 2008; Helin et al., 2018; Harrison et al., 2012; Zotter et al., 2017). The estimated average MACBC(880)biomass was 10.4 m2 g-1; this is more than twice the value for uncoated BC particles suggested by Bond and Bergstrom (2006) (MACBC(880)uncoated = 4.7 m2 g-1, extrapolated from 550nm to 880 nm by assuming AAEBC = 1.0)."

19. P10, Line 19, Line 85, the comma before '(2006)' should be deleted.

Response: We corrected this mistake:

"The estimated average MACBC(880)biomass was 10.4 m2 g-1; this is more than twice the value for uncoated BC particles suggested by Bond and Bergstrom (2006) (MACBC(880)uncoated = 4.7 m2 g-1, extrapolated from 550nm to 880 nm by assuming AAEBC = 1.0)."

20. P11, Line 5, 'may be relation with' should be changed to 'may be related with' or 'may have relation with'.

**Response:** We corrected this:

"Although unleaded gasoline has been used extensively in China since 2005, a considerable fraction of the Pb in the environment is still associated with vehicle-related particles, especially from the wear of metal alloys (Hao et al., 2019)."

21. P12, Line 26, 'This suggests and' should be changed to 'This suggests that'.

**Response:** We have changed the sentence as following:

"One can infer from this that biomass-burning emissions were responsible for the high BC-loading episode during the campaign."

22. P14, Line 12, 'the mainland China' should be changed to 'mainland China' or 'the mainland of China'.

**Response:** We have corrected this mistake in the revised manuscript:

"The air masses associated with Cluster 3 originated from the interior of China, and this group had the lowest BC mass concentrations of the three clusters,  $0.4 \pm 0.1 \,\mu g$  m-3. This third cluster composed small fraction of total trajectories (2%), and none of them were identified as polluted, suggesting minor influences from mainland China during the campaign."

"The transportation sector has grown rapidly in mainland China (Liu, 2019a), and the regional transport of motor vehicle emissions may have been the cause for the observed diurnal variations in  $BC_{fossil}$  for Cluster 3."

---

## Author Comment (AC2) · 29 Oct 2020

*Review of "Measurement report: Quantifying source contribution and radiative forcing of fossil fuel and biomass burning black carbon aerosol in the southeastern margin of Tibetan Plateau" (ACPD-2020-408, Liu et al)*

*General comment*

*This paper reports on measurements and modeling regarding the contribution of fossil fuel and biomass burning sources to black carbon (BC) aerosols abundance and radiative forcing at a site south-east of the Tibetan Plateau. Methods used in the study are robust, and the results are sound. However, it is difficult to ascertain the novelty and actual contribution to the overall understanding of, for instance, how BC aerosols are affecting the Tibetan Plateau. In my opinion, authors may turn this study into a relevant one if they would consider using the data at hand by better explaining the reasons that make these data important for improved understanding. In its present form, detailed measurements and modeling are more suited for a technical report not suitable, in my opinion, for this prestigious journal.*

**Response:** The authors appreciate the reviewer's valuable comments and suggestions, and we believe that the revised manuscript has been significantly improved after considering them. Below are point-to-point responses, and the modifications to the manuscript are marked.

*Specific comments*

*(1) The text would improve in clarity and possibly be shorten if reviewed by a native English writer/speaker. Also, some results could be summarized in tables improving the readability of the text.*

**Response:** We have had this manuscript polished by a native English speaking scientist. In addition, some results also are summarized in Table R1 (also see Table 1 in the revised manuscript) as suggested.

Table R1 Derived AAE, MAC and source contribution of BC from difference sources at the sampling site

|  | AAE | MAC ($m^2\ g^{-1}$) | Mass concentration ($\mu g\ m^{-3}$) | Contribution ratio |
|---|---|---|---|---|
| $BC_{biomass}$ | 1.7 | 10.4 | $0.4 \pm 0.3$ | 57% |
| $BC_{traffic}$ | 0.8 | 9.1 |  |  |
| $BC_{coal}$ | 1.1 | 15.5 |  |  |

| BC$_{fossil}$ | 0.9 | 12.3 | 0.3 ± 0.2 | 43% |
|---|---|---|---|---|

*(2) Abstract: Re-write according to suggestions below. What do we learn from this study? In what context is this useful? What is the novelty?*

**Response:** As suggested, we have rewritten the abstract and clearly state the purpose, importance and novelty of the study. It now reads as follows:

"Anthropogenic emissions of black carbon (BC) aerosol are transported from Southeast Asia to the southwestern Tibetan Plateau (TP) during the pre-monsoon; however, the quantities of BC from different anthropogenic sources and the transport mechanisms are still not well constrained because there have been no high-time-resolution BC source apportionments. Intensive measurements were taken in a transport channel for pollutants from Southeast Asia to the southeastern TP during the pre-monsoon to investigate the influences of fossil fuels and biomass burning on BC. A receptor model coupled multi-wavelength absorption with aerosol species concentrations was used to retrieve site-specific Ångström exponents (AAE) and mass absorption cross-sections (MAC) for BC. An 'aethalometer model' that used those values showed that biomass burning had a larger contribution to BC mass than fossil fuels (BC$_{biomass}$ = 57% versus BC$_{fossil}$ = 43%). The potential source contribution function indicated that BC$_{biomass}$ was transported to the site from northeastern India and northern Burma. The Weather Research and Forecasting model coupled with chemistry (WRF-Chem) model indicated that 40% of the BC$_{biomass}$ originated from Southeast Asia, while the highest BC$_{fossil}$ was transported from the southwest of the sampling site. A radiative transfer model indicated that the average atmospheric direct radiative effects (DRE) of BC were +4.6 ± 2.4 W m$^{-2}$ with +2.5 ± 1.8 W m$^{-2}$ from BC$_{biomass}$ and +2.1 ± 0.9 W m$^{-2}$ from BC$_{fossil}$. The DRE of BC$_{biomass}$ and BC$_{fossil}$ produced heating rates of 0.07 ± 0.05 and 0.06 ± 0.02 K day$^{-1}$, respectively. This study provides insights into sources of BC over a transport channel to the southeastern TP and the influence of the cross-border transportation of biomass burning emissions from Southeast Asia during the pre-monsoon."

• *Introduction*

*(3) Page 2, lines 13-14. In addition to characterizing source regions and their contributions to aerosol burden downwind it is also important to assess the timing in which this impact occur, how is the aging process, etc.*

**Response:** We agree with the reviewer that timing and aging are also importation factors, so we have re-written this sentence in the revised manuscript. It now reads as follows:

"Nonetheless, assessments of regional transport of bulk BC aerosol have not fully revealed the impacts of different BC emission sources because the optical properties and radiative effects of BC not only can vary among sources in complex ways but also can be affected by aging during transport (Tian et al., 2019; Zhang et al., 2019). Therefore, quantitative information on the contributions of different sources of BC over the TP is lacking, but it is critically needed for a better understanding the influence of anthropogenic emissions on its environment and climate."

*(4) Page 2, lines 22-24. Uncertainties in modeling studies not only depend on uncertain emission estimates but also on how well chemistry, transport and deposition processes are represented, initial/boundary conditions, etc. It appears necessary to review other studies to get an idea of the uncertainty when using models to simulate long-range transport, particularly over complex terrain.*

**Response:** We agree that the uncertainties of modeling method can be caused by the factors mentioned by reviewer and likely others. The complex terrain has impact on simulation of meteorological field over the Tibetan Plateau. We followed the reviewer's suggestion and have rewritten the relevant paragraph in the revised manuscript; it now reads as follows:

"On the other hand, the accuracy of model simulations is dependent on many factors, including uncertainties associated with initial particle parameters, aging processes, the accuracy of emission inventory, meteorological fields over the complex terrain, and the modules for chemistry and planetary boundary layer (PBL) dynamics, etc. (Koch et al., 2009; Madala et al., 2014; Vignati et al., 2010)."

*(5) Page 3, lines 1-24. This is a lengthy discussion about distinguishing between biomass and fossil fuel black carbon according to multiple observational and methodological studies. Rather than listing the pros and cons of the different methods, it would be good to have a clearer idea of which is the method fit for purpose to be discussed in the work. For that, it is key to establish a clear purpose, and how this will help improving understanding of a given phenomenon.*

**Response:** Following the reviewer's suggestion, we have rewritten the introduction in the revised manuscript and presented a clear explanation as to why we chose the online method, including why and how we optimized the methods. To clarify the purpose of this research, we re-organized the last two paragraphs of the introduction. They now read as follows:

"To make up for the deficiencies of filter-based analysis, BC source apportionments based on high-time resolution online data has been conducted in many locations but for the TP are limited. An 'aethalometer model' based on multi-wavelength absorption data is one of efficient approaches for distinguishing between BC from fossil fuel and biomass burning sources (Sandradewi et. al., 2008). Although it has been widely used elsewhere, this approach has not been applied to the TP. The accuracy of the 'aethalometer model' relies on the input parameters, including the Ångström exponents (AAE) and BC mass absorption cross-sections ($MAC_{BC}$) of different sources (Zotter et al., 2017). Limited information on site specific AAEs and $MAC_{BC}s$, lead most studies to rely on values taken from measurements made in other locations (e.g., Healy et al., 2017; Zhu et al., 2017). This results in unquantified uncertainties because the AAEs and $MAC_{BC}s$ vary with specific fuel subtypes and combustion conditions (Wang et al., 2018; Tian et al., 2019). Therefore, site-dependent AAE and $MAC_{BC}$ are essential for improving the reliability of BC source apportionment by the 'aethalometer model'.

In this study, field measurements of BC were taken on the southeastern margin of the TP during the pre-monsoon. This region connects the high altitude TP with the low altitude Yungui Plateau and forms a transport channel for pollutants from Southeast Asia (Wang et al., 2019a), and it is an ideal region for investigating the impact of pollutant transport to the southeastern TP. A receptor model that combined multi-wavelength absorption with aerosol species concentrations was used to retrieve site-dependent AAEs and $MAC_{BC}s$. This was done to improve the 'aethalometer model' with the goal of obtaining a more accurate BC source apportionment. The primary objectives of this study were to (1) quantify the mass concentrations of BC from fossil fuel and biomass burning sources; (2) determine the impact of regional transport on source-specific BC; and (3) assess the radiative effects caused by BC from different sources. This study provides insights into the BC sources on southeastern TP and an assessment of their radiative effects during the pre-monsoon."

*(6) Page 4, lines 1-6. You state that previous studies have dealt with radiative impacts of bulk BC, no distinguishing BC sources. Furthermore, you state that this study would be unique as it provides the first estimate of BC radiative forcing split by source regions. However, you estimate the instantaneous forcing over one site which is, by definition, locally representative, and not necessarily climatically important. Other studies may have estimated bulk BC forcing but over much longer periods of time, and over large areas, including the Himalayan cryosphere. Hence, I urge the authors to make their study unique by better establishing the purpose of it.*

**Response:** We appreciate this comment. Taking into consideration Comments 1–5, we have re-written the introduction to explain the purpose of our study and we deleted this paragraph in the

revised manuscript. BC is an important atmospheric light-absorbing material that can have significant radiative effects. The SBDART model has been widely used to estimate the instantaneous radiative effects of BC based on the ground observations (Gharibzadeh et al., 2017; Rajesh et al., 2018; Panicker et al., 2010). Although data from only one site on the southeastern TP were collected (because of practical limitations in personnel, equipment, logistics, etc.), we believe that the unique geographic location of sampling site (i.e., transport channel) on TP make our results of considerable interest. In addition, due to paucity of studies that have separately quantified BC mass from biomass burning and fossil fuels on TP, we think that it is important to understand their radiative effects and potential influences on climate.

**References:**

Gharibzadeh M , Alam K , Abedini Y , et al. Monthly and seasonal variations of aerosol optical properties and direct radiative forcing over Zanjan, Iran, J. Atoms. Sol-Terr. Phy.,164,268-275, dx.doi.org/10.1016/j.jastp.2017.09.006, 2017

Panicker, A. S., Pandithurai, G., Safai, P. D., Dipu, S., and Lee, D.-I.: On the contribution of black carbon to the composite aerosol radiative forcing over an urban environment, Atmos. Environ., 44, 3066-3070, 10.1016/j.atmosenv.2010.04.047, 2010.

Rajesh, T. A., and Ramachandran, S.: Black carbon aerosols over urban and high altitude remote regions: Characteristics and radiative implications, Atmos. Environ., 194, 110-122, 10.1016/j.atmosenv.2018.09.023, 2018.

• *Methodology*

*(7) Page 4, line 17. Improve the precision of attitude and longitude to allow a proper location of the site.*

**Response:** Following the reviewer's suggestion, we have specified the attitude and longitude of the sampling site in the revised manuscript. It now reads as follows:

"Intensive field measurements were made at the rooftop of a building (~10 m above the ground) at the Lijiang Astronomical Station, Chinese Academy of Sciences (3260 m above sea level, 100°1'48"E, 26°41'24"N), Gaomeigu County, Yunnan Province, China (Fig. 1) from 14 March to 13 May 2018. (Fig. 1)."

*(8) Page 4, lines 18-19. As per your reference, Wang et al (2019a), your observation site is located*

*along a "transportation channel". Describe the overall transport patterns affect. Is the period of observations representative of which transport/circulation pattern? An overall meteorological description is missing.*

**Response:** In the pre-monsoon season when the southeastern margin of the TP is influenced by the westerly winds (Chan et al 2017; Niu et al., 2017), a pathway for the cross border transport of emissions from southeast Asia to the TP. The sampling period for this study was from March to May and therefore in the pre-monsoon. In the revised manuscript, we have added some information about the 'transport channel'. It now reads as follows:

> "During the campaign, westerly winds created a potential pathway for cross border transport from southeast Asia to southwest China. During the study, the average relative humidity and temperature were $80\% \pm 20\%$ and $7.6 \pm 3.2\,℃$, respectively; the mean wind speed near surface was $5.4 \pm 2.1$ m s$^{-1}$, and the winds were mainly from the west and southwest."

**References:**

Chan, C. Y., Wong, K. H., Li, Y. S., Chan, Y., and Zhang, X. D.: The effects of Southeast Asia fire activities on tropospheric ozone, trace gases and aerosols at a remote site over the Tibetan Plateau of Southwest China, Tellus B , 58B, 310-318, 10.1111/j.1600-0889.2006.00187.x, 2017

Niu H., Kang., S., Zhang, Y., Shi, X., Shi., X., Wang S., Li, G., Yan, X., Pu, T. He, Y., Distribution of light-absorbing impurities in snow of glacier on Mt. Yulong, southeastern Tibetan Plateau, Atmos. Res., 197, 474-484, 10.1016/j.atmosres.2017.07.004, 2017.

*(9) Page 4, lines 19-20. You say that the population surrounding your observational site is small. Small compared to what? Then you go onto establishing that limited anthropogenic activities are found there. However, your results show a non-negligible contribution. The site should be better described, including a brief description of aerosol sources.*

**Response:** At the sampling site, the influence of anthropogenic activities is limited due to the low population density and lack of industries. The local emissions have small effects on the BC source apportionment results compared with the contributions of fossil fuel BC from two highways (5.5 km from the sampling site) and transport from the border with Burma. Following the reviewer's suggestion, we have added some information about the possible anthropogenic emissions in the surrounding area of the sampling site. The revised manuscript now reads as follows:

"The sampling site is 3–5 km from Gaomeigu village, which has 27 households and 110 residents. Villagers there rely on farming for their livelihoods, and biomass is the primarily residential fuel (Li et al, 2016). There are no large industries near the village and traffic is light. However, two highways (Hangzhou-Ruili Expressway and Dali-Nujiang Expressway) are located ~5.5 km to the west of the sampling site."

*(10)    Page 7, section 2.5. HYSPLIT can be used with large-scale (synoptic) meteorological fields. Do you have an assessment of how well this approach works over complex terrain?*

**Response:** The meteorological data used in this study were obtained from the Global Data Assimilation System (GDAS) and had a spatial resolution of $1° \times 1°$. The HYSPLIT model converted the vertical layers from the original coordinate system into its own terrain-following coordinate system (sigma) and directly used the data contained in meteorological files for the trajectory calculations (Draxler and Hess, 1997). The surface in the terrain-following coordinate system is consistent with the ground, and that solves the problem of modelling near mountainous areas (Phillips, 1965). Furthermore, this method has been used over complex terrain with various meteorological data in a number of studies (Burley and Bytnerowicz 2011; Wang et al., 2015; Wang et al.,2019; Qu et al 2015 and Khan et al., 2010).

To determine if the trajectory would be impacted by the surface rising, we also ran calculations at heights of 150m and 1000m in addition to 500m (Figure R1). The results showed that directions were similar, particularly between the results at150m and at 500m. We finally decided to use the 500m results because greater heights be higher than the height at which the samples were collected, and 500m is generally representative of the average planetary boundary height at the site (~600m).

[Figure]

**Figure R1.** the blue lines represent the clusters at 500m height, the yellow lines represent the clusters at 150m height, the red lines represent the cluster at 1000m height.

**References:**

Burley J D , Bytnerowicz A . Surface ozone in the White Mountains of California, Atmospheric Environment, 45, 4591-4602, 10.1016/j.atmosenv.2011.05.062, 2011.

Chengkai Qu, Xinli Xing, Stefano Albanese, et al. Spatial and seasonal variations of atmospheric organochlorine pesticides along the plain-mountain transect in central China: Regional source vs. long-range transport and air‑soil exchange, Atmos. Environ. 122, 31-40, 10.1016/j.atmosenv.2015.09.008, 2015.

Draxler, R., and Hess, G.: An overview of the HYSPLIT_4 modelling system for trajectories, Aust. Meteorol. Mag., 47, 1998.

Khan A J , Li J , Dutkiewicz V A , et al. Elemental carbon and sulfate aerosols over a rural mountain site in the northeastern United States: Regional emissions and implications for climate change, Atmos. Environ., 44, 2364-2371, 10.1016/j.atmosenv.2010.03.025, 2010.

Phillips N.A. a coordinate system having some special advantages for numerical forecasting, Shorter contributions, J. Atmos. Sci., 14, 184-185, 1957.

Wang, Q. Y., Huang, R. J., Cao, J. J., Tie, X. X., Ni, H. Y., Zhou, Y. Q., Han, Y. M., Hu, T. F.,

Zhu, C. S., Feng, T., Li, N., and Li, J. D.: Black carbon aerosol in winter northeastern Qinghai–Tibetan Plateau, China: the source, mixing state and optical property, Atmos. Chem. Phys., 15, 13059-13069, 10.5194/acp-15-13059-2015, 2015.

Wang, Q., Han, Y., Ye, J., Liu, S., Pongpiachan, S., Zhang, N., Han, Y., Tian, J., Wu, C., Long, X., Zhang, Q., Zhang, W., Zhao, Z., and Cao, J.: High Contribution of Secondary Brown Carbon to Aerosol Light Absorption in the Southeastern Margin of Tibetan Plateau, Geophys. Res. Lett., 46, 4962-4970, 10.1029/2019gl082731, 2019.

*(11)    Why do you choose 3-day back trajectories instead of 2 or 5 days?*

**Response:** Studies have indicated that BC lifetime varies from 3.3 to 12 days in the atmosphere (Liu et al., 2011 and references therein). The BC lifetime depends on many factors such as morphology, size, mixing state, aging condition, and meteorological conditions. It is not possible to know the exact lifetime of the BC sampled at the study site, and therefore, we chose the lowest value for the BC lifetimes to minimize the effects of BC deposition during transport to the site. In addition, the 3-day backward trajectories also have been widely used in previous studies to investigate BC transport pathways (e.g., Wang et al., 2018; Verma et al., 2010).

**References:**

Liu, J., Fan, S., Horowitz, W. L., Levy, H., 116, Evaluation of factors controlling long‐range transport of black carbon to the Arctic, J. Geophys. Res., doi:10.1029/2010JD015145, 2011.

Verma, R. L., Sahu L. K., Kondo, Y., Takegawa, N., Han, S., Jung, J. S., Kim, Y., J., Fan, S., Sugimoto, N., Shammaa, M. H., Zhang, Y., H., and Zhao, Y.: Temporal variations of black carbon in Guangzhou, China, in summer 2006, Atmos. Chem. Phys., 10, 6471–6485, 10.5194/acp-10-6471-2010, 2010.

Wang, Q., Cao, J., Han, Y., Tian, J., Zhu, C., Zhang, Y., Zhang, N., Shen, Z., Ni, H., Zhao, S., and Wu, J.: Sources and physicochemical characteristics of black carbon aerosol from the southeastern Tibetan Plateau: internal mixing enhances light absorption, Atmos. Chem. Phys., 18, 4639-4656, 10.5194/acp-18-4639-2018, 2018.

*• Results and discussion*

*(12)    Page 11, lines 20-25. Your BC aerosol appears to have aged. Can't you use your WRF-*

*Chem simulations to attempt providing further insights on this issue?*

**Response:** We thank the reviewer providing us with this suggestion. However, BC is usually considered as chemically inert in the atmosphere (Bond et al., 2013), and the WRF-Chem model only accounts for physical processes of BC in the atmosphere, such as the wet and dry deposition. Although we concluded that BC underwent substantial aging, the objective of the study was to apportion the BC sources, and that did not include contributions from materials coating the BC. Moreover, due to the limitation of the measurement methods in this study, we could not obtain quantitative information regarding BC aging, which would have been the best way to constrain the model simulation. Therefore, the aging of BC is something that would be better addressed in future studies.

**References:**

Bond, T. C., Doherty, S. J., Fahey, D. W., Forster, P. M., Berntsen, T., DeAngelo, B. J., Flanner, M. G., Ghan, S., Kärcher, B., Koch, D., Kinne, S., Kondo, Y., Quinn, P. K., Sarofim, M. C., Schultz, M. G., Schulz, M., Venkataraman, C., Zhang, H., Zhang, S., Bellouin, N., Guttikunda, S. K., Hopke, P. K., Jacobson, M. Z., Kaiser, J. W., Klimont, Z., Lohmann, U., Schwarz, J. P., Shindell, D., Storelvmo, T., Warren, S. G., and Zender, C. S.: Bounding the role of black carbon in the climate system: A scientific assessment, J. Geophys. Res.-Atmos., 118, 5380-5552, 10.1002/jgrd.50171, 2013.

*o Page 12, section 3.2.*

*(13)    Some of your results could be better appreciated if summarized in a table.*

**Response:** As suggested, results have been summarized in to a table. Please see the response of comment 1 above.

*(14)    You make multiple references to FigS3. Maybe it is better to bring it to the main manuscript. If so, it could be useful to split the graphs for daytime and nighttime periods as it would better fit with Figure 2.*

**Response:** The mass concentrations of levoglucosan and benzothiazolone were obtained from 24 h filter samples, and so we cannot compare daytime versus nighttime periods. Note that the online

BC data were integrated to match each filter sampling times. We did follow the reviewer's suggestion and combined Fig. S3 with Fig. 2 in the revised manuscript (also see Fig. R2 below).

[Figure]

**Figure R2.** Scatter plots of (a) biomass burning BC (BC$_{biomass}$) versus fossil fuel combustion BC (BC$_{fossil}$), (b) BC$_{biomass}$ versus levoglucosan, and (c) BC$_{fossil}$ versus benzothiazolone. BC$_{biomass}$ and BC$_{fossil}$ represent black carbon aerosol contributed by biomass burning and fossil fuel sources, respectively. (d)Time series of hourly averaged mass concentrations of black carbon (BC) aerosol from biomass bunirng (BC$_{biomass}$) and fossil fuel sources (BC$_{fossil}$) during the campaign.

• *Conclusion*

*(15)    Stress the novelty, and make it explicit that the period studied correspond to a given set of transport/circulation patterns.*

**Response:** We followed the reviewer's suggestion and rewrote the conclusions in the revised manuscript. It now reads as follows:

"This study quantified the source contributions of BC aerosol from fossil fuel and biomass burning at a site on the southeastern margin of the TP that represents a regional transport channel for air pollution. The study was conducted during pre-monsoon when the

southeastern TP was heavily influenced by the air mass from southeast Asia. To reduce the uncertainties caused by interferences in absorption measurements (i.e. secondary absorption and dust) and assumptions relative to AAE and $MAC_{BC}$, the traditional 'aethalometer model' was optimized in two aspects. First, a BC-tracer method coupled with a minimum R-squared approach was applied to separate secondary absorption from the total absorption, and as a result, the interferences of absorption from secondary aerosols have been eliminated. Then, an optical source apportionment model that used primary multi-wavelength absorption and chemical species as inputs was used to derive site-dependent AAE and $MAC_{BC}$ values— these minimize the uncertainties associated with prior assumptions on these parameters. The AAE ($MAC_{BC}$) calculated in this way was 0.9 (12.3 $m^2$ $g^{-1}$) for the fossil fuel source and 1.7 (10.4 $m^2$ $g^{-1}$) for biomass burning. The results of 'aethalometer model' that used these values showed that the average mass concentration of BC was 0.7 ± 0.5 μg $m^{-3}$ of which 57% was from biomass burning and 43% from fossil fuels. Trajectory analysis showed that the $BC_{biomass}$ over the site was mainly driven by regional transport from northeastern India and Burma, while $BC_{fossil}$ was primarily influenced by traffic emissions from areas surrounding the sampling site. Moreover, the WRF-Chem model indicated that biomass burning in Southeast Asia contributed 40% of the BC loadings over the southeastern margin of the TP. The SBDART model showed that a DRE of +4.6 ± 2.4 W $m^{-2}$ for the total $PM_{2.5}$ BC, of which +2.5 ± 1.8 W $m^{-2}$ was from $BC_{biomass}$ and +2.1 ± 0.9 W $m^{-2}$ from $BC_{fossil}$. The results of our study provide useful information concerning the sources of BC over an atmospheric transport channel to the southeastern TP, and they highlight the importance of the cross-border transport of biomass burning emissions from Southeast Asia on the region during the pre-monsoon."

---

## Author Response (AR1)

***Anonymous Referee #1***

*General comments: In this manuscript, the sources of BC aerosols over the Tibetan Plateau and their radiative effects were investigated. BC aerosols were distinguished into fossil fuel combustion source and biomass burning source. Regional transport of source-specific BC was further explored by models. On this basis, the radiative effects caused by BC from different sources were evaluated. Overall, the manuscript is well structured, the methods are technically sound, and the main findings presented seem to be reasonable and be of general interests to the Tibetan Plateau ecosystem and climate stability study. I think the topic fits within the scope of ACP. I would recommend acceptance of this manuscript for publication pending the following revisions:*

**Response:** We sincerely thank the reviewer for the comments and suggestions, and we have revised the relevant text and content. Below are point-to-point responses, and the modifications to the manuscript are marked.

***Specific comments:***

*1.Please spend time picking through the manuscript and check for spelling and grammatical errors, especially the tense, prepositions and articles. For example, 'BC on the TP' should be replaced with 'BC over the TP', 'transport to TP' should be replaced with 'transport to the TP'*

**Response:** We have taken the suggestion to heart and have corrected the relevant mistakes as shown below. Also, the paper has been polished by a native English speaker.

"Therefore, quantitative information on the contributions of different sources of BC over the TP is lacking, but it is critically needed for a better understanding the influence of anthropogenic emissions on its environment and climate."

"Nonetheless, these studies have been helpful for understanding the sources of BC over the TP."

*2.Section 1, this part should introduce the research background and significance, current status, concealed problems, as well as research mentality and content of this study. Please highlight the innovation and importance from another angle and reduce describing the deficiencies of previous research appropriately.*

**Response:** We have re-written the introduction to include more background material, to clarify the research focus, to explain why we were interested in study area and why we chose the methods we did. The revised introduction now reads:

[revised manuscript text omitted]

*3.Please try to avoid expressions like 'our study', which seems not be objective. Technical corrections: 1.P1, Line 24, 'reveal' should be changed to 'revealed'.*

**Response:** We have changed all "our study" in the manuscript into "this study". The verbal tenses have been correct in the following sentence:

"The potential source contribution function indicated that $BC_{biomass}$ was transported to the site from northeastern India and northern Burma."

*2.P1, Line 26, add 'which' before 'can explain'.*

**Response:** We have changed the sentence to

"The Weather Research and Forecasting model coupled with chemistry (WRF-Chem) model indicated that 40% of the $BC_{biomass}$ originated from Southeast Asia"

*3.P2, Line 1, delete 'and' before 'heating rates of'.*

**Response:** We have changed this sentence to

"The DRE of $BC_{biomass}$ and $BC_{fossil}$ produced heating rates of $0.07 \pm 0.05$ and $0.06 \pm 0.02$ K day$^{-1}$, respectively."

*4.P2, Line 1, 'The glaciers on the TP recently shows are rapidly retreating' should be revised.*

**Response:** We have corrected this to

"The glaciers on the TP are facing rapid retreat, however, and if unchecked, that could result in adverse effects on Asian hydrological cycle and the Asian monsoon (Luo et al., 2020; Hua et al., 2019)."

*5.P2, Line 4-8, Please cite these literatures, doi:10.1093/nsr/nwz191, doi.org/10.1016/j.atmosenv.2020.117583, doi: 10.1016/j.atmosenv.2019.04.001.*

**Response:** We cite those papers in the revised version.

"The Tibetan Plateau (TP) is an important regulator of climate change in the northern hemisphere, and it plays crucial roles in the functions of global ecosystems and climate stability (Liu et al., 2019b; Liu et al., 2020a). The TP is covered by one of the largest ice masses on Earth, and it has been called the water tower of Asia (Liu et al., 2020b)."

*6.P2, Line 8-10, Please cite these literatures, doi.org/10.3390/rs12020231, doi:10.1002/joc.6430.*

**Response:** We now cited these references in the revised version.

"The glaciers on the TP are facing rapid retreat, however, and if unchecked, that could result in adverse effects on Asian hydrological cycle and the Asian monsoon (Luo et al., 2020; Hua et al., 2019)."

*7.P2, Line 16, 'the atmospheric BC studies on the TP' should be changed to 'studies on the TP atmospheric BC'.*

**Response:** We have revised the sentence

"In addition, numerous studies have shown that the high bulk BC mass loadings and the associated regional influences on the TP are related to transport from South Asia (Liu et al., 2015; Han et al., 2020; Cong et al., 2015; Wang et al., 2015)."

*8.P2, Line 16-18, Please cite the literature, doi: 10.5194/acp-15-12581-2015.*

**Response:** We have read that paper and now cite it. Please see it in the answer for the above question.

*9.P2, Line 24, 'the other is based on the field observations to apportion BC into different sources through a certain data analytical method.' Please add appropriate literature.*

**Response:** We have re-written this text, and now it reads as follows:

"Several studies have assessed the contributions of different BC sources through model simulations or isotopic methods. For example, Zhang et al. (2015) investigated BC sources for different parts of the TP by using a chemical transport model and a source tagging approach, and they found that the contributions of BC sources varied among regions and with the seasons. Li et al., (2016) used filter sampling and carbon isotopes ($\Delta^{14}C$ and $\delta^{13}C$) to determine the BC from fossil fuels and biomass burning in several areas of the TP."

*10.P2, Line 25, 'transport to TP' should be changed to 'transport to the TP'.*

**Response:** We have corrected and add the article "the" into TP in the revised version.

*11.P3, Line 5, 'are advantageous to capturing' should be changed to 'are advantageous to capture'.*

**Response:** We revised the sentence

"A major disadvantage of filter-based measurements is they are constrained by low time resolution, and this makes it challenging to capture the detailed evolution of pollution events."

12. P3, Line 10, 'cross-section (MAC) used in the model.' Please add appropriate literature.

**Response:** We have added the relevant literature:

"The accuracy of the 'aethalometer model' relies on the input parameters, including the Ångström exponents (AAE) and BC mass absorption cross-sections ($MAC_{BC}$) of different sources (Zotter et al., 2017)."

13. P4, Line 11, 'on the rooftop of' should be changed to 'at on the rooftop of'.

**Response:** We have corrected "on" into "at". The sentence now reads

"Intensive field measurements were made at the rooftop of a building (~10 m above the ground) at the Lijiang Astronomical Station, Chinese Academy of Sciences (3260 m above sea level, 100°1'48"E, 26°41'24"N), Gaomeigu County, Yunnan Province, China (Fig. 1) from 14 March to 13 May 2018."

14. P4, Line 15, 'the radiative effect' should be changed to 'the radiative effects'.

**Response:** We have changed all "radiative effect" into "radiative effects" in the revised version.

15. P5, Line 4, 'was resolved using' should be changed to 'was resolved by using'.

**Response:** We have changed the sentence as below:

"A dual-spot technique for the aethalometer measurements was used to compensate for non-linearity, while a factor of 2.14 was used to correct for the artifacts caused by the quartz filters (Drinovec et al., 2015)."

16. P8, Line 10, 'the number of the endpoints' should be changed to 'the number of endpoints'.

**Response:** We have deleted "the" before endpoints in the sentence:

"where $m_{ij}$ is the number of endpoints associated with BC mass concentration higher than the set criterion;"

17. P9, Line 8, 'model is elaborated in Ricchiazzi and Yang, (1998)' should be changed to 'model was elaborated in Ricchiazzi and Yang (1998)'.

**Response:** We have corrected the tense as follows:

"The direct radiative effects (DRE) of source-specific BC were estimated with the widely used Santa Barbara DISORT Atmospheric Radiative Transfer (SBDART) model—a detailed description of which may be found in Ricchiazzi and Yang (1998)."

*18. P10, Line 15, delete 'which is' before 'within a relative boarder range'. P10, Line 18, delete 'which was' before 'over two times'. In addition, it is necessary to pay attention to the tense errors, which often appear throughout the manuscript.*

**Response:** Thanks for pointing out the error in tense. The paper has been edited and thoroughly revised, and the problems with tense should have been corrected. The sentences were changed to

"That was within the relatively broad range of $AAE_{biomass}$ (1.2–3.5) determined by other methods (e.g., $\triangle^{14}C$ and organic tracers) in previous studies (Sandradewi et al., 2008; Helin et al., 2018; Harrison et al., 2012; Zotter et al., 2017). The estimated average $MAC_{BC}(880)_{biomass}$ was 10.4 $m^2$ $g^{-1}$; this is more than twice the value for uncoated BC particles suggested by Bond and Bergstrom (2006) ($MAC_{BC}(880)_{uncoated} = 4.7$ $m^2$ $g^{-1}$, extrapolated from 550nm to 880 nm by assuming $AAE_{BC} = 1.0$)."

*19. P10, Line 19, Line 85, the comma before '(2006)' should be deleted.*

**Response:** We corrected this mistake:

"The estimated average $MAC_{BC}(880)_{biomass}$ was 10.4 $m^2$ $g^{-1}$; this is more than twice the value for uncoated BC particles suggested by Bond and Bergstrom (2006) ($MAC_{BC}(880)_{uncoated} = 4.7$ $m^2$ $g^{-1}$, extrapolated from 550nm to 880 nm by assuming $AAE_{BC} = 1.0$)."

*20. P11, Line 5, 'may be relation with' should be changed to 'may be related with' or 'may have relation with'.*

**Response:** We corrected this:

"Although unleaded gasoline has been used extensively in China since 2005, a considerable portion of the Pb in the environment is still associated with vehicle-related particles, especially from the wear of metal alloys (Hao et al., 2019)."

*21. P12, Line 26, 'This suggests and' should be changed to 'This suggests that'.*

**Response:** We have changed the sentence as following:

"One can infer from this that biomass-burning emissions were responsible for the high BC loading episode during the campaign."

*22. P14, Line 12, 'the mainland China' should be changed to 'mainland China' or 'the mainland of China'.*

**Response:** We have corrected this mistake in the revised manuscript:

"The air masses associated with Cluster 3 originated from the interior of China, and this group had the lowest BC mass concentrations of the three clusters, $0.4 \pm 0.1$ µg m$^{-3}$. This third cluster composed small fraction of total trajectories (2%), and none of them were identified as polluted, suggesting minor influences from mainland China during the campaign."

"The transportation sector has grown rapidly in mainland China (Liu, 2019a), and the regional transport of motor vehicle emissions may have been the cause for the observed diurnal variations in BC$_{fossil}$ for Cluster 3."

*Review of "Measurement report: Quantifying source contribution and radiative forcing of fossil fuel and biomass burning black carbon aerosol in the southeastern margin of Tibetan Plateau" (ACPD-2020-408, Liu et al)*

*Anonymous Referee #2*

*General comment: This paper reports on measurements and modeling regarding the contribution of fossil fuel and biomass burning sources to black carbon (BC) aerosols abundance and radiative forcing at a site south-east of the Tibetan Plateau. Methods used in the study are robust, and the results are sound. However, it is difficult to ascertain the novelty and actual contribution to the overall understanding of, for instance, how BC aerosols are affecting the Tibetan Plateau. In my opinion, authors may turn this study into a relevant one if they would consider using the data at hand by better explaining the reasons that make these data important for improved understanding. In its present form, detailed measurements and modeling are more suited for a technical report not suitable, in my opinion, for this prestigious journal.*

**Response:** The authors appreciate the reviewer's valuable comments and suggestions, and we believe that the revised manuscript has been significantly improved after considering them. Below are point-to-point responses, and the modifications to the manuscript are marked.

***Specific comments***

*(1) The text would improve in clarity and possibly be shorten if reviewed by a native English writer/speaker. Also, some results could be summarized in tables improving the readability of the text.*

**Response:** We have had this manuscript polished by a native English speaking scientist. In addition, some results also are summarized in Table R1 (also see Table 1 in the revised manuscript) as suggested.

Table 1 Derived Ångström absorption exponents (AAE), Mass absorption coefficients (MAC) and percent source contribution of black carbon (BC) from difference sources

| | AAE | MAC (m$^2$ g$^{-1}$) | Mass concentration (µg m$^{-3}$) | Contribution ratio |
|---|---|---|---|---|
| BC$_{biomass}$ | 1.7 | 10.4 | 0.4 ± 0.3 | 57% |
| BC$_{traffic}$ | 0.8 | 9.1 | --- | --- |
| BC$_{coal}$ | 1.1 | 15.5 | --- | --- |
| BC$_{fossil}$ | 0.9 | 12.3 | 0.3 ± 0.2 | 43% |

*(2) Abstract: Re-write according to suggestions below. What do we learn from this study? In what context is this useful? What is the novelty?*

**Response:** As suggested, we have rewritten the abstract and clearly state the purpose, importance and novelty of the study. It now reads as follows:

"Anthropogenic emissions of black carbon (BC) aerosol are transported from Southeast Asia to the southwestern Tibetan Plateau (TP) during the pre-monsoon; however, the quantities of BC from different anthropogenic sources and the transport mechanisms are still not well constrained because there have been no high-time-resolution BC source apportionments. Intensive measurements were taken in a transport channel for pollutants from Southeast Asia to the southeastern TP during the pre-monsoon to investigate the influences of fossil fuels and biomass burning on BC. A receptor model coupled multi-wavelength absorption with aerosol species concentrations was used to retrieve site-specific Ångström exponents (AAE) and mass absorption cross-sections (MAC) for BC. An 'aethalometer model' that used those values showed that biomass burning had a larger contribution to BC mass than fossil fuels (BC$_{biomass}$ = 57% versus BC$_{fossil}$ = 43%). The potential source contribution function indicated that BC$_{biomass}$ was transported to the site from northeastern India and northern Burma. The Weather Research and Forecasting model coupled with chemistry (WRF-Chem) model indicated that 40% of the BC$_{biomass}$ originated from Southeast Asia, while the highest BC$_{fossil}$ was transported from the southwest of the sampling site. A radiative transfer model indicated that the average atmospheric direct radiative effects (DRE) of BC were +4.6 ± 2.4 W m$^{-2}$ with +2.5 ± 1.8 W m$^{-2}$ from BC$_{biomass}$ and +2.1 ± 0.9 W m$^{-2}$ from BC$_{fossil}$. The DRE of BC$_{biomass}$ and BC$_{fossil}$ produced heating rates of 0.07 ± 0.05 and 0.06 ± 0.02 K day$^{-1}$, respectively. This study provides insights into sources of BC over a transport channel to the southeastern TP and the influence of the cross-border transportation of biomass burning emissions from Southeast Asia during the pre-monsoon."

*• Introduction*

*(3) Page 2, lines 13-14. In addition to characterizing source regions and their contributions to aerosol burden downwind it is also important to assess the timing in which this impact occur, how is the aging process, etc.*

**Response:** We agree with the reviewer that timing and aging are also importation factors, so we have re-written this sentence in the revised manuscript. It now reads as follows:

"Nonetheless, assessments of regional transport of bulk BC aerosol have not fully revealed the impacts of different BC emission sources because the optical properties and radiative effects of BC not only can vary among sources in complex ways but also can be affected by aging during transport (Tian et al., 2019; Zhang et al., 2019). Therefore, quantitative information on the contributions of different sources of BC over the TP is lacking, but it is critically needed for a better understanding the influence of anthropogenic emissions on its environment and climate."

*(4) Page 2, lines 22-24. Uncertainties in modeling studies not only depend on uncertain emission estimates but also on how well chemistry, transport and deposition processes are represented, initial/boundary conditions, etc. It appears necessary to review other studies to get an idea of the uncertainty when using models to simulate long-range transport, particularly over complex terrain.*

**Response:** We agree that the uncertainties of modeling method can be caused by the factors mentioned by reviewer and likely others. The complex terrain has impact on simulation of meteorological field over the Tibetan Plateau. We followed the reviewer's suggestion and have rewritten the relevant paragraph in the revised manuscript; it now reads as follows:

"On the other hand, the accuracy of model simulations is dependent on many factors, including uncertainties associated with initial particle parameters, aging processes, the accuracy of emission inventory, meteorological fields over the complex terrain, and the modules for chemistry and planetary boundary layer (PBL) dynamics, etc. (Koch et al., 2009; Madala et al., 2014; Vignati et al., 2010)."

*(5) Page 3, lines 1-24. This is a lengthy discussion about distinguishing between biomass and fossil fuel black carbon according to multiple observational and methodological studies. Rather than listing the pros and cons of the different methods, it would be good to have a clearer idea of which is the method fit for purpose to be discussed in the work. For that, it is key to establish a clear purpose, and how this will help improving*

*understanding of a given phenomenon.*

**Response:** Following the reviewer's suggestion, we have rewritten the introduction in the revised manuscript and presented a clear explanation as to why we chose the online method, including why and how we optimized the methods. To clarify the purpose of this research, we re-organized the last two paragraphs of the introduction. They now read as follows:

"To make up for the deficiencies of filter-based analysis, BC source apportionments based on high-time resolution online data has been conducted in many locations but for the TP are limited. An 'aethalometer model' based on multi-wavelength absorption data is one of efficient approaches for distinguishing between BC from fossil fuel and biomass burning sources (Sandradewi et. al., 2008). Although it has been widely used elsewhere, this approach has not been applied to the TP. The accuracy of the 'aethalometer model' relies on the input parameters, including the Ångström exponents (AAE) and BC mass absorption cross-sections (MAC$_{BC}$) of different sources (Zotter et al., 2017). Limited information on site specific AAEs and MAC$_{BCS}$, lead most studies to rely on values taken from measurements made in other locations (e.g., Healy et al., 2017; Zhu et al., 2017). This results in unquantified uncertainties because the AAEs and MAC$_{BCS}$ vary with specific fuel subtypes and combustion conditions (Wang et al., 2018; Tian et al., 2019). Therefore, site-dependent AAE and MAC$_{BC}$ are essential for improving the reliability of BC source apportionment by the 'aethalometer model'.

In this study, field measurements of BC were taken on the southeastern margin of the TP during the pre-monsoon. This region connects the high altitude TP with the low altitude Yungui Plateau and forms a transport channel for pollutants from Southeast Asia (Wang et al., 2019a), and it is an ideal region for investigating the impact of pollutant transport to the southeastern TP. A receptor model that combined multi-wavelength absorption with aerosol species concentrations was used to retrieve site-dependent AAEs and MAC$_{BCS}$. This was done to improve the 'aethalometer model' with the goal of obtaining a more accurate BC source apportionment. The primary objectives of this study were to (1) quantify the mass concentrations of BC from fossil fuel and biomass burning sources; (2) determine the impact of regional transport on source-specific BC; and (3) assess the radiative effects caused by BC from different sources. This study provides insights into the BC sources on southeastern TP and an assessment of their radiative effects during the pre-monsoon."

*(6) Page 4, lines 1-6. You state that previous studies have dealt with radiative impacts of bulk BC, no distinguishing BC sources. Furthermore, you state that this study would*

*be unique as it provides the first estimate of BC radiative forcing split by source regions. However, you estimate the instantaneous forcing over one site which is, by definition, locally representative, and not necessarily climatically important. Other studies may have estimated bulk BC forcing but over much longer periods of time, and over large areas, including the Himalayan cryosphere. Hence, I urge the authors to make their study unique by better establishing the purpose of it.*

**Response:** We appreciate this comment. Taking into consideration Comments 1–5, we have re-written the introduction to explain the purpose of our study and we deleted this paragraph in the revised manuscript. BC is an important atmospheric light-absorbing material that can have significant radiative effects. The SBDART model has been widely used to estimate the instantaneous radiative effects of BC based on the ground observations (Gharibzadeh et al., 2017; Rajesh et al., 2018; Panicker et al., 2010). Although data from only one site on the southeastern TP were collected (because of practical limitations in personnel, equipment, logistics, etc.), we believe that the unique geographic location of sampling site (i.e., transport channel) on TP make our results of considerable interest. In addition, due to paucity of studies that have separately quantified BC mass from biomass burning and fossil fuels on TP, we think that it is important to understand their radiative effects and potential influences on climate.

*(10) Page 7, section 2.5. HYSPLIT can be used with large-scale (synoptic) meteorological fields. Do you have an assessment of how well this approach works over complex terrain?*

**Response:** The meteorological data used in this study were obtained from the Global Data Assimilation System (GDAS) and had a spatial resolution of $1°×1°$. The HYSPLIT model converted the vertical layers from the original coordinate system into its own terrain-following coordinate system (sigma) and directly used the data contained in meteorological files for the trajectory calculations (Draxler and Hess, 1997). The surface in the terrain-following coordinate system is consistent with the ground, and that solves the problem of modelling near mountainous areas (Phillips, 1965). Furthermore, this method has been used over complex terrain with various meteorological data in a number of studies (Burley and Bytnerowicz 2011; Wang et al., 2015; Wang et al.,2019; Qu et al 2015 and Khan et al., 2010).

To determine if the trajectory would be impacted by the surface rising, we also ran calculations at heights of 150m and 1000m in addition to 500m (Figure R1). The results showed that directions were similar, particularly between the results at150m and at 500m. We finally decided to use the 500m results because greater heights be higher than the height at which the samples were collected, and 500m is generally representative of the average planetary boundary height at the site (~600m).

[Figure]

**Figure R1.** the blue lines represent the clusters at 500m height, the yellow lines represent the clusters at 150m height, the red lines represent the cluster at 1000m height.

*○ Page 12, section 3.2.*

*(13)    Some of your results could be better appreciated if summarized in a table.*

**Response:** As suggested, results have been summarized in to a table. Please see the response of comment 1 above.

*(14)    You make multiple references to FigS3. Maybe it is better to bring it to the main manuscript. If so, it could be useful to split the graphs for daytime and nighttime periods as it would better fit with Figure 2.*

**Response:** The mass concentrations of levoglucosan and benzothiazolone were obtained from 24 h filter samples, and so we cannot compare daytime versus nighttime periods. Note that the online BC data were integrated to match each filter sampling times. We did follow the reviewer's suggestion and combined Fig. S3 with Fig. 2 in the revised manuscript (also see Fig. R2 below).

[Figure]

**Figure R2.** Scatter plots of (a) biomass burning BC ($BC_{biomass}$) versus fossil fuel combustion BC ($BC_{fossil}$), (b) $BC_{biomass}$ versus levoglucosan, and (c) $BC_{fossil}$ versus benzothiazolone. $BC_{biomass}$ and $BC_{fossil}$ represent black carbon aerosol contributed by biomass burning and fossil fuel sources, respectively. (d)Time series of hourly averaged mass concentrations of black carbon (BC) aerosol from biomass bunirng ($BC_{biomass}$) and fossil fuel sources ($BC_{fossil}$).

• *Conclusion*

*(15)    Stress the novelty, and make it explicit that the period studied correspond to a given set of transport/circulation patterns.*

**Response:** We followed the reviewer's suggestion and rewrote the conclusions in the revised manuscript. It now reads as follows:

[revised manuscript text omitted]
…Bb…nzothiazolone is a substance …eleased from the breaking-…own of the antioxidant in motor vehiclevehicular…tires (Cheng et al., 2006) while….Br is anothera…tracer of closely related to …otor vehicle emission (Guo et al., 2009). Similarly,, and…Zn and Cu are associated with the combustion of lubricating fluids and the wearing…of brakes and tires (Lough et al., 2005; Song et al., 2006). FinallyMeanwhile… EC and OC also are can be …omponents ofused to denote…motor vehicle emissions (Cao et al., 2013). Although unleaded gasoline has been used extensively used in China since 2005, a considerable portion of Pb in the environment is still associated with found in …ehicle-related particles, especially from which may be relation with…the wear of metal alloys (Hao et al., 2019). Therefore, this second source factor was identified as traffic-related emissions. This source constitutes a moderate percentage of primary $b_{abs}(\lambda)$ (15–30%). The estimated traffic emission-related AAE ($AAE_{traffic}$) was 0.8, consistent with the findingfeature…that BC is the dominant light-absorbing carbonaceous aerosol speciescarbon…for traffic emissions (Kirchstetter et al., 2004). The $AAE_{traffic}$ found here is …lso was close to a value obtained using thethose obtained with…$\triangle^{14}C$ approach (Zotter et al., 2017). The estimated BC

设置了格式: 下标

删除了: is a…typically associated withfeature of…coal combustion (Hsu et al., 2016; Kim and Hopke, 2008). Although coal is nota…usedscarce…extensively near the site energy used …n the TP, emissions from it can be influenced by…coal combustion may have been transported to site from surrounding areas (e.g., East Asia, Li et al., 2016). This source contributed only …2–17% of primary $b_{abs}(\lambda)$, which is less than that from biomass burning or and …raffic emissions. The obtained AAE of coal combustion ($AAE_{coal}$ = 1.1) was similar to the $AAE_{traffic}$, suggesting that BC was also thea…dominant light-absorbing carbon specie in coal combustion emissions. The $AAE_{coal}$ wasis…close to the value of chunk coal combustion (1.3) but lower than that for to

删除了: , which may …eflects the coal …ypes of coal transported to that affected BC particles at …aomeigu, at least to some degree. The estimated BC MAC(880) of coal combustion ($MAC_{BC}(880)_{coal}$ = 15.5 $m^2$ $g^{-1}$) was larger than $MAC_{BC}(880)_{biomass}$ and $MAC_{BC}(880)_{traffic}$. Numerous studies have confirmed that aged BC could result in MAC increased by a factor of 1.5 – 3.5 relative to uncoated one (Chen et al., 2017; Ma et al., 2020). …he enhance factor for $MAC_{BC}(880)_{coal}$ (3.3) falls near the upper limit of this range noted above,…and although tT…is is likely large enhancement should be …elated to the aging processes …f BC particles during their long-range …ransport Gaomeigu, although…
[revised manuscript text omitted]

删除了: the southeastern TP… AsSince…there were s…minimal impacts fromcarce…traffic activities …t night, the $BC_{fossil}$ loadings remained at a…steady concentration …rom 21:00 to 08:00. The stable nocturnal $BC_{fossil}$ may reflect the impact of fossil fuel emissions on BC in the southeastern margin of the TP due to the accumulation resulting from effect driven by

删除了: 4…). Cluster #… originated from northeastern India and then passed over Bangladesh before arriving Gaomeigu. The average BC mass concentration of this cluster was the highest $(0.8 \pm 0.4 \ \mu g \ m^{-3})$ of among …he three clusters. About 74% of total trajectories were associated with Cluster #…, of which 22% was identified as polluted and had ones with …n average BC loading of $1.3 \pm 0.5 \ \mu g \ m^{-3}$. Cluster #… originated overderived from…Burma hadwith…an average BC loading of $0.7 \pm 0.7 \ \mu g \ m^{-3}$. This cluster c accounted foronstituted…only 24% of total trajectories, but among them, about 37% was referred to pollution with BC reaching as high as $1.6 \pm 0.9 \ \mu g \ m^{-3}$. The air masses associated with Cluster #… originated from the interior of mainland…China, and which…this group had the lowest BC mass concentrations of the three clusters,of…$0.4 \pm 0.1 \ \mu g \ m^{-3}$. This third cluster composedcomprised…small fraction of total trajectories (2%), and none of them wereas…identified as pollutedpollution…

删除了: different…clusters were further …sed to investigate the impacts of regional transport. As shown in Fig. 43… and e, a …imilar diurnal variations of …n $BC_{biomass}$ was …ere found for Clusters #… and Cluster…#…—, …hey had bothwith…larger values during daytime (8:00–12:00) compared with the …ight-time… This pattern of higher daytime $BC_{biomass}$ was associated withThe enhancements of daytime $BC_{biomass}$ (8:00 – 12:00) were related to…the …egional transport from northeastern India (Cluster #…) and Burma (Cluster #…). For Cluster #…, the diurnal variation of…$BC_{biomass}$ decreased during the daytime…and increased at night (Fig. 43…), and that pattern trackedwhich was mainly controlled by…the daily variationsvariability…in PBL height. Unlike Clusters 1 and 2,Compared to other two clusters,…the inverse …iurnal variation of $BC_{biomass}$ for Cluster #… were more likely due toindicates the…influences of biomass-burning activities from areasthe…surrounding the sampling site  areas rather …han regional transport. However, it should be noted that these cases were uncommonscarce…because of only 2% of air-masses were associated with Cluster #

删除了: a …imilar diurnal patterns was …ere found for Clusters #… and Cluster #… (Fig. 43… and e), most likely due to which was mainly associated with …he influences of traffic emissions from in…surrounding areas as well as daily cycles of PBL height as discussed in section 3.2. The $BC_{fossil}$ loadings of Cluster #… (Fig.3g…g) wereexhibited a

sporadic fluctuations. Unlike the declining trend of $BC_{fossil}$ during the daytime found for Clusters 1 and 2, the relative stable $BC_{fossil}$ loadings in Cluster 3 indicate that there were emissions from fossil fuel sources that offset the effect of the changes in PBL height. The transportation sector has grown rapidly in mainland China (Liu, 2019), and the regional transport of motor vehicle emissions may have been the

5 cause for the observed diurnal variations in $BC_{fossil}$ for Cluster 3.

The PSCF model was applied to further investigate the likely spatial distribution of pollution source regions for $BC_{biomass}$ and $BC_{fossil}$. As shown in Fig. 5b, a low PSCF value of $BC_{biomass}$ was found near Gaomeigu while high values were concentrated in the northeastern India and northern Burma, consistent with intensive fire activities in these areas (Fig. S4). This indicates that large $BC_{biomass}$ loadings at

10 Gaomeigu were more likely influenced by cross-border transport of biomass burning rather than local emissions. For $BC_{fossil}$ (Fig. 5c), the most likely impact region was located to the southwest of Gaomeigu, near where two highways are located (e.g., Hangzhou-Ruili Expressway and Dali-Nujiang Expressway). Owing to the low consumption of coal in the southeastern TP (Li et al., 2016), the high PSCF values of $BC_{fossil}$ were more likely from traffic emissions than coal combustion. Moreover, sporadic high PSCF

15 values of $BC_{fossil}$ also were found in the northern Burma, indicating possible influences of fossil fuel emissions here.

To further quantify the contributions of the BC transported from Southeast Asia to Gaomeigu, we studied a high BC episode (23 – 27 March) using a simulation with WRF-Chem model. Two scenarios of emissions were simulated: one involved all BC emission sources, and the other turned off biomass-

20 burning emissions in Southeast Asia. The variation of modelled BC mass concentration shows an acceptable degree of consistency with the measured values (r = 0.63, $p < 0.01$, Fig. S5), Furthermore, the index of agreement was estimated to be 0.77, indicating that the development of this BC episode was effectively captured by the WRF-Chem model. Nonetheless, the normalized mean bias between the measured and modelled BC values was estimated to be 24%, suggesting that simulation was biased

25 towards high values. This discrepancy is best attributed to the uncertainties in the simulation associated with the emission inventory and meteorological conditions. Fig. 6a shows the spatial distributions of BC loadings in Gaomeigu and surrounding areas. The mass concentrations of BC at times exceeded 15 μg m$^{-3}$ over Burma and northern India, and that is more than an order of magnitude higher compared with the

删除了： diurnal variation expect for …poradic fluctuations. Unlike the declining trend of $BC_{fossil}$ during the daytime found for Clusters 1 and 2in the other two clusters… the relative stable $BC_{fossil}$ loadings variation…in Cluster #… indicates…that there were emissions from fossil fuel sources that to …ffset the effect of the changes inpollutant diffusion caused by increased…PBL height. The transportation sector has grown rapidlyOwing to the high-density transportation network…in mainland China (Liu, 2019), and the regional transport of motor vehicletraffic…emissions may have been the cause for the observed diurnal variations in $BC_{fossil}$ for Cluster 3be the cause

删除了： Further,… t…e PSCF model was applied to further investigate the likely spatial distribution of pollution source regions for $BC_{biomass}$ and $BC_{fossil}$. As shown in Fig. 54…, a low PSCF value of $BC_{biomass}$ was found near Gaomeigu while high values were concentrated in the northeastern India and northern Burma, consistent with intensive fire activitiesspots…in these areas (Fig. S46…. This indicates that large $BC_{biomass}$ loadings at Gaomeigu were more likely influenced by cross-border transport of biomass burning rather than local emissions. For $BC_{fossil}$ (Fig. 54…), the most likelypossible…impact region was located to near …he southwest of Gaomeigu, near where two highways are locatedthere are a few villages scattered around and two highways…(e.g., Hangzhou-Ruili Expressway and Dali-Nujiang Expressway). Owing to the low consumption of coal in the southeastern TP (Li et al., 2016), the high PSCF values of $BC_{fossil}$ were more likely frommay be mainly contributed by…traffic emissions rather …han coal combustion. Moreover, sporadic high PSCF values of $BC_{fossil}$ also were also

删除了： transported…to BC at …aomeigu, we studied a high BC episode (23 – 27 March) using a simulationwas arbitrary selected and…withsimulated by the

删除了： one …r = 0.63, $p < 0.01$, Fig. S57…, Furthermore, and …he index of agreement was estimated to be 0.77, indicating that the development formation process…of this BC episode was effectively captured by the WRF-Chem model. NonethelessHowever… it should be noted that …he normalized mean bias between the measured and modelled BC values was estimated to be 24%, suggesting that simulation was biased towards high valuesan overestimation of simulation… This discrepancy is best attributedcould be attributed…to the simulation

删除了： resulted from…the emission inventory and meteorological conditions. Fig. 65… shows the spatial distributions of BC loadings in Gaomeigu and surrounding areas. The mass concentrations of BC at timescan…exceeded 15 μg m$^{-3}$ overin…Burma and northern India, and that iswhich was…more thanover…one …n order of magnitude higher compared withthan that in

[revised manuscript text omitted]

删除了: regions
删除了: 9
删除了: T
删除了: the
删除了: is a
删除了: T
删除了: mass
删除了: explored
删除了: and the direct radiative forcing
删除了: the black carbon (
删除了: )
删除了: combustion
删除了: in the
删除了:
删除了: Tibetan Plateau
删除了: The observed mean BC concentration was 0.7 ± 0.5 µg m$^{-3}$ during the campaign. Based on the optical source apportionment using multi-wavelength absorption and chemical species, the obtained absorption Ångström exponents (AAEs) and BC mass absorption cross section (MAC) at wavelength of 880 nm were
删除了: and
删除了: and
删除了: , respectively
删除了: Using these source-specific AAEs and BC MACs, the improved aethalometer model estimated
删除了: source
删除了: ($BC_{fossil}$ and $BC_{biomass}$, respectively)
删除了: The diurnal cycle in $BC_{biomass}$ was driven by BC regional
删除了: (
删除了: )
设置了格式: 下标
删除了: from the
删除了: can
删除了: at the sampling site
删除了: The southeast of Gaomeigu was the most likely contribut
删除了: Santa Barbara DISORT Atmospheric Radiative Transfer (
删除了: )
删除了: estimated
删除了: a cooling effect of -3.0 ± 1.5 W m$^{-2}$ at the Earth's surface
删除了: ,

[revised manuscript text omitted]

删除了: 5

删除了: were

[Figure]

删除了:

删除了: 6

删除了: F

[Figure]

**Figure 7.** The average direct radiative forcing (DRE) of black carbon (BC) aerosol and PM2.5 at the Earth's surface (SUF), the top of atmsphere (TOA), and in the atmosphere (ATM = TOA – SUF). The BC_biomass and BC_fossil represent BC contributed by biomass burning and fossil fuel sources, respectively.

---

## Referee Report (RR1)

**Second review of "Measurement report: Quantifying source contribution and radiative forcing of fossil fuel and biomass burning black carbon aerosol in the southeastern margin of Tibetan Plateau" (ACPD-2020-408, Liu et al)**

**General comment**

The authors have significantly improved the manuscript. The different sections are easier to follow. Also, they have highlighted the novelty and contribution of this work. In other words, authors have turned this study into a relevant as it provides data and results that are important for improved understanding of anthropogenic influences on the cryosphere of the Tibetan Plateau. Thus, except for minor changes, I recommend acceptance.

**Minor changes to be introduced**

- The text should be revised in terms of punctuation. There are several places that need correction. For instance, on line 17 of the abstract it says "apportionments..", should say "apportionments." In other places, references are quoted as Alberts et al., (YEAR). It should say Alberts et al. (YEAR). All in all, check the text for typos.
- In the introduction, the authors cite Bond et al (2013) and Ming et al (2009) as references indicating the role played by BC in accelerating cryosphere retreat. I urge the authors to, in addition, review more up to date references. Consider for instance:

    o Zhang, R., Wang, H., Qian, Y., Rasch, P. J., Easter, R. C., Ma, P. L., et al. (2015). Quantifying sources, transport, deposition, and radiative forcing of black carbon over the Himalayas and Tibetan Plateau. *Atmos. Chem. Phys.* 15, 6205–6223. doi:10.5194/acp-15-6205-2015.
    o Xu, Y., Ramanathan, V., and Washington, W. M. (2016). Observed high-altitude warming and snow cover retreat over Tibet and the Himalayas enhanced by black carbon aerosols. Atmos. Chem. Phys. 16, 1303–1315. doi:10.5194/acp-16-1303-2016.
    o Gertler, C. G., Puppala, S. P., Panday, A., Stumm, D., and Shea, J. (2016). Black carbon and the Himalayan cryosphere: A review. Atmos. Environ. 125, 404–417. doi:10.1016/j.atmosenv.2015.08.078.
    o Kang, S., Zhang, Y., Qian, Y., and Wang, H. (2020). A review of black carbon in snow and ice and its impact on the cryosphere. Earth-Science Rev. 210. doi:10.1016/j.earscirev.2020.103346.
    o Yoshida, A., Moteki, N., Ohata, S., Mori, T., Koike, M., Kondo, Y., et al. (2020). Abundances and Microphysical Properties of Light-Absorbing Iron Oxide and Black Carbon Aerosols Over East Asia and the Arctic. J. Geophys. Res. Atmos. 125. doi:10.1029/2019JD032301.

- Introduction, line 22. You state "numerous" studies but you refer to just a few. I suggest you replace "numerous" by "several".

---

## Referee Report (RR2)

The authors have substantially and satisfactory revised the manuscript according to the reviewer's comments. This revised manuscript can be accepted for publication.